# A multi-center, international, randomized, 2-year, parallel-group study to assess the superiority of IVUS-guided PCI versus qualitative angio-guided PCI in unprotected left main coronary artery (ULMCA) disease: Study protocol for OPTIMAL trial

Giovanni Luigi De Maria[1], Luca Testa[2], Jose M. de la Torre Hernandez[3], Dimitrios Terentes-Printzios[1], Maria Emfietzoglou[1], Roberto Scarsini[1], Francesco Bedogni[2], Ernest Spitzer[4,5], Adrian Banning[1] *

1 Heart Centre, John Radcliffe Hospital, Oxford University Hospitals, NHS Foundation Trust, Oxford, United Kingdom, 2 Coronary Revascularisation Unit, IRCCS Policlinico S. Donato, San Donato Milanese, Milan, Italy, 3 Cardiology Department, Hospital Universitario Marques de Valdecilla, IDIVAL, Santander, Spain, 4 European Cardiovascular Research Institute, Rotterdam, The Netherlands, 5 Department of Cardiology, Thoraxcenter, Erasmus Medical Centre, Rotterdam, The Netherlands

* Adrian.Banning@ouh.nhs.uk

## Abstract

### Background

Percutaneous coronary intervention (PCI) is used increasingly for revascularization of unprotected left main coronary artery (LMCA) disease. Observational studies and subgroup analyses from clinical trials, have suggested a possible benefit from the use of intravascular ultrasound (IVUS) guidance when performing unprotected LMCA PCI. However, the value of imaging with IVUS has never been proven in an appropriately powered randomized clinical trial. The OPtimizaTIon of Left MAin PCI With IntravascuLar Ultrasound (OPTIMAL) trial has been designed to establish whether IVUS-guided PCI optimization on LMCA is associated with superior clinical outcomes when compared with standard qualitative angiography-guided PCI.

### Methods

The OPTIMAL trial is a randomized, multicenter, international study designed to enroll a total of 800 patients undergoing PCI for unprotected LMCA disease. Patients will be randomized in a 1:1 fashion to IVUS-guided PCI versus angiogram-guided PCI. In patients allocated to the angiogram-guided arm, use of IVUS is discouraged, unless there are safety concerns. In patients allocated to the IVUS guidance arm, pre-procedural IVUS assessment is highly recommended, whilst post-procedural IVUS assessment is mandatory to confirm appropriate stenting result and/or to guide stent result optimization, according to predefined criteria. Patients will be followed up to 2 years after the index procedure. The primary

**Funding:** Yes. This is an Investigator-initiated study under the umbrella of the ECRI. Research grants have been provided by Boston Scientific and Philips Volcano to fund this study. ECRI maintains clinical trial insurance coverage for study participants in the event of trial-related injuries, if applicable and in accordance with the applicable laws and regulations of the country in which the study is performed. Also, according to the applicable regulatory requirement(s), ECRI provides insurance or indemnity (legal and financial coverage) to the Investigator/the Institution against claims arising from the trial, except for claims that arise from malpractice and/or negligence. The funders had no role in study design, data collection and analysis, decision to publish, or preparation of the manuscript.

**Competing interests:** Dr De Maria reports speaker fees from Miracor Medical SA and research grants from Abbott and Philips. Dr Testa reports fees as medical proctor for Boston Scientific, Meril, Concept Medical, Abbott, Philips and advisory board member and/or speaker fees and/or institutional research grant from Boston Scientific, Philips, Abbott, Medtronic, Terumo, Concept Medical. Dr de la Torre Hernandez reports receipt of grants/research supports from Abbott Medical, Biosensors, Bristol Myers Squibb, Amgen and receipt of honoraria or consultation fees from Boston Scientific, Medtronic, Biotronik, Astra Zeneca, Daiichi-Sankyo. Dr Bedogni reports fees as medical proctor for BSCI, Meril, Medtronic, Terumo and advisory board member and/or speaker fees and/or institutional research grant from Boston Scientific, Philips, Abbott, Medtronic, Terumo, Concept Medical. Prof Banning reports institutional grant for fellowship form Boston and speaker fees Boston, Phillips and Miracor Medical SA. Dr Spitzer declares that the sponsor of the study is the European Cardiovascular Research Institute (ECRI), in which he is a board member, and the research organization executing the study is Cardialysis, in which he is the chief medical officer. Dr Scarsini, Dr Terentes-Printzios, Dr Emfietzoglou have no conflict of interest to declare. In specific relationship with the study funders: • Boston Scientific has provided research grant support to Dr Testa, Dr Bedogni and Prof Banning • Boston Scientific has provided speaker fees/ consultancy fees/proctor fees to Dr Testa, Dr de la Torre Hernandez, Dr Bedogni and Prof Banning • Philips has provided research grant support to Dr De Maria, Dr Testa and Dr Bedogni • Philips has provided speaker fees/consultancy fees/proctor fees to Dr Testa, Dr Bedogni and Prof Banning We confirm that: "This does not alter investigators'

outcome measure is the Academic Research Consortium (ARC) patient-oriented composite endpoint (PoCE) which includes all-cause death, any stroke, any myocardial infarction and any repeat revascularization at 2 years follow-up.

## Discussion

The OPTIMAL trial aims to provide definitive evidence about the clinical impact of IVUS-guidance during PCI to an unprotected LMCA. It is anticipated by the investigators, that an IVUS-guided strategy will be associated with less clinical events compared to a strategy guided by angiogram alone.

## Trial registration

ClinicalTrials.gov: NCT04111770. Registered on October 1, 2019.

## Introduction

Atherosclerotic involvement of the LMCA is relatively unusual and it is observed in 4% of all coronary angiograms [1]. However, since LMCA supplies at least two thirds of the left ventricle and since atheroma in the LMCA involves the distal bifurcation in most cases (up to 80%), disease of the LMCA poses significant challenges both from the prognostic and technical point of view [2]. Data from dedicated randomized clinical trials have highlighted the potential of coronary stenting as mode of revascularization for unprotected LMCA, especially in patients with low to moderate degree of complexity of coronary anatomy (Anatomical SYNTAX Score $\leq$ 32) or in patients unsuitable for cardiac surgery [3–5]. This equipoise in terms of long-term mortality between percutaneous coronary intervention (PCI) and coronary artery bypass grafting (CABG) in unprotected LMCA disease has been recently confirmed in two large independent meta-analyses [6, 7] and by the ten-years follow-up results of the SYNTAX [8] and PRECOMBAT studies [9].

In order to guarantee a durable result of PCI in LMCA, that is able to match the proven outcomes from cardiac surgery, PCI has to be appropriately and meticulously planned, and the result has to be optimized. Because of its intrinsic technical limitations, angiography has a suboptimal performance in assessing the LMCA. Angiography allows a limited assessment of the extension of disease around the LMCA bifurcation and of the overall atherosclerotic burden, especially in case of eccentric distribution of the atherosclerotic plaque. Additionally, it cannot offer any insight about vessel wall characteristics, such as vessel remodeling phenomenon or plaque composition, especially in terms of true calcium content [10]. Similarly, coronary angiography is limited in assessing appropriate stent expansion, apposition or adequate lesion coverage. In contrast to two-dimensional angiography, IVUS is able to assess both the lumen and the features of the arterial wall, and when compared to optical coherence tomography (OCT), IVUS has much higher tissue penetration [11].

### Role of intravascular ultrasound in assessing the LMCA

**Defining severity of stenosis.** When there is an intermediate stenosis of the LMCA demonstrated by angiography, a functional assessment (with either fractional flow reserve [FFR], or instantaneous free-wave ratio [iFR]) is typically the recommended approach. However, IVUS assessment of minimal lumen area (MLA) on LMCA can also be used. This reflects the fact that the LMCA represents the area where the anatomical assessment of a coronary

stenosis, based on intravascular imaging, correlates the best with functional evaluation. The study by Jasti et al was the first to validate a LMCA MLA of 6 mm$^2$ against FFR [12], and subsequently, the LITRO study confirmed, against clinical outcomes, the safety in deferring PCI in LMCA stenosis with an MLA above 6 mm$^2$ [13]. For this reason, when decision about revascularization on angiographic intermediate LMCA stenosis is based on IVUS, the consensus document from EuroBifurcation Club, proposes to use a cut off of MLA $> 6$ mm$^2$ for safely deferring LMCA intervention [14].

**Defining anatomy, "normal" references and atheroma distribution in LMCA.** IVUS dimensions for a normal LMCA (or a minimally diseased LMCA) and the subtended left anterior descending (LAD) or left circumflex (LCx) are remarkably consistent [15]. Whilst coronary angiography cannot account for positive remodeling, IVUS can identify proximal and distal reference sites free (or relatively free) from atheroma both in the LMCA as in the LAD and LCx by providing an accurate and easy definition of both lumen area and external elastic lamina (EEL) area. As a result, IVUS can favor optimal stent sizing and accurate selection of stent landing zones (areas with plaque burden $< 50\%$) [11]. Understanding the actual distribution of atheroma around the distal LMCA bifurcation has clearly implications in terms of selection of bifurcation-stenting technique. A pivotal study on 140 patients with distal LMCA atherosclerosis has elegantly reported how LMCA bifurcation disease is rarely focal and that, irrespective of the angiographic appearance, atheroma extends from the LMCA into the proximal LAD in 90% of cases, from the LMCA into the LCx artery in 66% and from the LMCA into both the LAD and LCx arteries in 62% of cases [16]. In practice, a provisional crossover approach should be considered initially whenever possible. However, when there is IVUS evidence of a large atheroma burden ($> 60\%$) and/or MLA $< 4$ mm$^2$ at the ostium of the side branch (usually the LCx), this finding is associated with a high chance of side branch ostium compromise after a provisional cross-over approach [17], and these data suggest an "upfront" two stent strategy should be considered.

**Stent optimization.** In the SYNTAX II study, IVUS adoption triggered actions for stent-optimization, including further stenting or postdilation, in 30.2% of cases [18]. More specifically for the LMCA setting, the EXCEL study reported IVUS triggered actions for stent-optimization in 51.7% of cases (further postdilation in 29% of cases, use of larger balloons in 30% of cases and higher inflation pressures in 17% of cases) [4]. Kang et al have assessed the impact of stent expansion, as assessed by IVUS, on the future risk of stent failure in unprotected LMCA PCI. Specifically, they identified thresholds of minimum stent area (MSA) at the ostium of LAD and LCx, at the polygon of confluence (POC) and in the LMCA associated with the risk of future in-stent restenosis. These so called "Kang's IVUS criteria" are proposed as minimal MSA targets to achieve in order to address stent underexpansion in LMCA bifurcation. Specifically, stent optimization (e.g., either post-dilations, kissing balloon inflation, further stenting) should be performed until the MSA of the proximal LMCA, polygon of convergence, ostial LAD and the ostial LCx are at least 8, 7, 6 and 5 mm$^2$, respectively [19]. Whilst Kang's criteria remain the ones recommended by the EuroBifurcation Club Guidelines [20], it must be acknowledged that they have mainly been tested in Asian patients who are known to have smaller sized LMCA ($5.2\pm1.8$ vs. $6.2\pm1.4$ mm$^2$; $p<0.0001$) [21].

## Rationale

IVUS is undoubtedly useful in providing thorough assessment of the LMCA and currently has a class IIa B indication for both evaluation of LMCA disease and guidance of stent-optimization according to European Guidelines [22]. However, this recommendation is mainly based on evidence coming from observational studies or sub-analysis of trials comparing the effectiveness of PCI in LMCA versus CABG [22].

Specifically, the sub-analysis of the MAIN-COMPARE registry, including 402 propensity-matched patients undergone PCI to LMCA, has been one of the first reports suggesting a potential clinical benefit of IVUS-guidance in LMCA PCI. In the overall cohort (combining patients treated with drug eluting stent [DES] and bare metal stent [BMS]), IVUS guidance was not associated with a superior clinical outcome. However, when the analysis was restricted to 290 patients treated with DES, IVUS-guidance offered a clinical benefit in terms of lower risk of cumulative death and MI. Interestingly, this benefit was only evident after the second year of follow-up and was not mirrored by a concomitant lower rate of target vessel revascularization, which did not differ significantly between the angio-guided and IVUS-guided group [23]. Subsequently, the IVUS-TRONCO-ICP study reported the superior benefit of IVUS-guidance in terms of cumulative cardiac death, MI, and target lesion revascularization in a larger cohort of 1010 propensity-matched patients undergone LMCA PCI. Notably, this study highlighted the potential clinical impact of IVUS-guidance in reducing stent thrombosis, especially in PCI to distal LMCA and in cases treated with double stent techniques [24].

Despite consistent signals coming from these reports, the fact that this evidence is derived from retrospective, non-randomized studies remains a significant limitation. Additionally, these studies did not directly test the role of IVUS-guidance as the main objective and the decision about performing IVUS was usually left to operator's discretion. Also, these limitations affected the recent sub-analysis of IVUS data from NOBLE trial, which showed how IVUS adoption was associated with lower rate of target lesion revascularization on LMCA, but it was not related with a lower rate of major adverse cardiac or cerebrovascular events [25]. In addition, these studies did not contemplate consistent and standardized IVUS protocol, or criteria to guide stent implantation and optimization. Notably, the benefits of a specific IVUS stent optimization protocol have recently been demonstrated to reduce the rate of future LMCA interventions [26].

Moreover, it should be noted that early data from initial single-centre studies have actually suggested that high-volume operators usually achieve better outcome in LMCA PCI, minimizing the additional benefit that IVUS assessment could provide [27]. But despite the fact that the use of IVUS has increased, it is still not ubiquitous and even in the most recent trials on LMCA PCI such as EXCEL or NOBLE, IVUS guidance was reported in the range of 77.2% and 74.9% of cases respectively [3, 4]. Therefore, a dedicated randomized study assessing the clinical impact of IVUS-guidance in optimization of LMCA-PCI is desirable. The Optimization of left main percutaneous coronary intervention with intravascular ultrasound randomized controlled trial (OPTIMAL study) has been designed to ascertain the superiority of IVUS-guided PCI optimization on unprotected LMCA compared to angiography-guidance. Administrative information about the OPTIMAL trial is described in Tables 1 and 2. Despite its limitations, qualitative angio-guided PCI is still the standard treatment for patients with LMCA disease. Its selection as comparator is therefore justified.

## Objectives

The objective of the OPTIMAL trial is to assess the superiority of an IVUS-guided approach versus a qualitative angio-guided approach in optimizing PCI result on unprotected LMCA. Our hypothesis is that an IVUS-guided strategy will be associated with less clinical events compared to an angio-guided strategy.

## Materials and methods

### Trial design

The OPTIMAL trial is a randomized, controlled, multicenter, international, post-marketing, superiority, parallel group study. A total of 800 participants will be enrolled and randomized in a 1:1 fashion to IVUS-guided PCI versus qualitative angio-guided PCI, stratified by site.

**Table 1. Administrative information for OPTIMAL trial.**

| Title [2] | OPtimizaTIon of Left MAin PCI With IntravascuLar Ultrasound. The OPTIMAL Randomized Controlled Trial |
|---|---|
| Trial registration {2a and 2b}. | ClinicalTrials.gov: NCT04111770 |
| | See **Table 2** for all items from WHO Trial Registration Dataset. |
| Protocol version {3} | Issue Date: May 3, 2021 |
| | Protocol Amendment Number: 2.0 (final) |
| | Author(s): Maaike Alkema (Cardialysis, B.V.) |
| | Revision chronology: |
| | • Version 1.0 (13 November 2019) |
| | • Version 2.0 (final) (3 May 2021) |
| Funding {4} | The OPTIMAL trial is an investigator-initiated study sponsored by the European Cardiovascular Research Institute (ECRI, Rotterdam, The Netherlands), and funded by Boston Scientific Corporate (Marlborough, Massachusetts, USA) and Philips Volcano (Zaventem, Belgium). |
| Author details {5a} | • Adrian Banning MD, MBBS: Heart Centre, John Radcliffe Hospital, Oxford University Hospitals, NHS Foundation Trust, Oxford, United Kingdom |
| | • Luca Testa MD: Coronary Revascularisation Unit, IRCCS Policlinico S. Donato, San Donato M.ne, Milan, Italy |
| | • Giovanni Luigi De Maria MD, PhD: Heart Centre, John Radcliffe Hospital, Oxford University Hospitals, NHS Foundation Trust, Oxford, United Kingdom |
| | • Francesco Bedogni MD: Coronary Revascularisation Unit, IRCCS Policlinico S. Donato, San Donato M.ne, Milan, Italy |
| | • Jose M de la Torre Hernandez MD, PhD: Cardiology Department., Hospital Universitario Marques de Valdecilla, IDIVAL, Santander, Spain |
| | • Ernest Spitzer, MD: European Cardiovascular Research Institute, Rotterdam, The Netherlands |
| | • AB is the Study Chairman and the Study Principal Investigator. LT is the Deputy Chairman and the Study Principal Investigator. GDM is the Country Lead in UK. FB is the Country Lead in Italy. JTH is the Country Lead in Spain. ES is the Sponsor Representative. |
| Name and contact information for trial sponsor {5b} | Trial Sponsor: ECRI-13 |
| | Contact name: Ernest Spitzer, MD |
| | Address: ECRI-13 b.v., Westblaak 98, Rotterdam, 3012 KM, Netherlands |
| | Telephone: +31 (0)10 206 28 00 |
| | Email: e.spitzer@ecri-trials.com |
| Role of sponsor {5c} | ECRI acts as the trial sponsor and runs the project management of the study. ECRI has outsourced trial execution to Cardialysis BV (Rotterdam, The Netherlands), which is an independent Contract Research Organization (CRO). Cardialysis participated in the study design and is responsible for performing all trial operations including project and site management, data management, statistical analysis, and safety reporting. |
| | IVUS assessment will be performed with either Opticross HD system (Boston Scientific), or Refinity, Revolution or Eagle Eye Platinum systems (Philips Volcano). Boston Scientific and Philips Volcano had no role in the design of this study and will not have any role during its execution, analysis, data interpretation, or decision to submit results. |

Both the investigators and the participants will not be blinded for the procedure. All patients will be followed up to 2 years after the index procedure. Total study duration from first patient in to last patient out is expected to be approximately 4 years (2 years enrollment, 2 years follow-up) (Figs 1 and 2).

**Table 2. World Health Organization trial registration data set.** {2b}

| Data category | Information |
|---|---|
| Primary registry and trial identifying number | ClinicalTrials.gov |
| | NCT04111770 |
| Date of registration in primary registry | September 30, 2019 |
| Secondary identifying numbers | ECRI-013 |
| Source(s) of monetary or material support | Boston Scientific (Marlborough, Massachusetts, USA) |
| | Philips Volcano (Zaventem, Belgium) |
| Primary sponsor | European Cardiovascular Research Institute (ECRI, Rotterdam, The Netherlands) |
| Secondary sponsor(s) | Not applicable |
| Contact for public or scientific queries | European Cardiovascular Research Institute |
| | Ernest Spitzer, MD |
| | ECRI-13 b.v., Westblaak 98, Rotterdam, 3012 KM, Netherlands |
| | +311102062802 |
| | e.spitzer@ecri-trials.com |
| Public title | The OPTIMAL Randomized Controlled Trial (OPTIMAL) |
| Scientific title | OPtimizaTIon of Left MAin PCI With IntravascuLar Ultrasound. The OPTIMAL Randomized Controlled Trial |
| Countries of recruitment | Italy |
| | United Kingdom |
| | Spain |
| Health condition(s) or problem(s) studied | Left Main Coronary Artery Stenosis |
| Intervention(s) | • Investigational strategy: IVUS-guided approach in the setting of left main PCI |
| | • Reference strategy: qualitative angio-guided approach in the setting of left-main PCI |
| Key inclusion and exclusion criteria | Inclusion Criteria |
| | 1. The patient must be ≥18 years of age |
| | 2. De novo lesion in an unprotected left main coronary artery (ULMCA; ostial, shaft or distal); OR ostial left anterior descending artery (LAD) or ostial circumflex artery (LCX), both compatible with one Medina class of LM disease; or ostial intermediate branch disease |
| | 3. PCI is considered appropriate and feasible by the treating interventionalist |
| | 4. Silent ischemia, stable angina, unstable angina, or non-ST-segment elevation MI (NSTEMI) |
| | Able to understand and provide informed consent and comply with all study procedures, including follow-up for at least 2 years |
| | Note: A patient with a prior CABG with no patent bypass on the left main coronary artery (LMCA) can be included. |
| | Exclusion criteria: |
| | 1. Patient is a woman who is pregnant or nursing |
| | 2. Female patient of childbearing potential, i.e., who are not surgically sterile or post-menopausal (defined as no menses for 2 years without an alternative cause) |
| | 3. IVUS is strictly required for pre-PCI lesion severity assessment |
| | 4. ST-elevation myocardial infarction, cardiogenic shock |
| | 5. Previous history of CABG with patent graft to the LAD and/or patent graft to the LCX |
| | 6. Prior PCI of the LM, ostial LAD or ostial LCX at any time prior to enrollment |
| | 7. Prior PCI of any other (i.e., non-LM, non-ostial-LAD and non-ostial-LCX) coronary artery lesions within 30 days prior to enrollment |
| | 8. Patients unable to tolerate, obtain or comply with dual antiplatelet therapy for at least 6 months in stable patients and 1 year in ACS patients |
| | 9. Known contraindication or hypersensitivity to everolimus, platinum-chromium, or to anticoagulants |
| | 10. Patients requiring additional surgery (cardiac or non-cardiac) within 3 months post-enrollment |
| | 11. Non-cardiac co-morbidities with a life expectancy less than 2 years |
| | 12. Currently participating in another trial that is not yet at its primary outcome; the patient is not allowed to participate in another investigational device or drug study for at least 12 months after enrollment |
| Study type | Interventional |
| | Allocation: randomized |
| | Intervention model: parallel assignment |
| | Masking: None (open label) |
| | Primary purpose: Treatment |
| Date of first enrolment | July 19, 2020 |
| Target sample size | 800 |

*(Continued)*

**Table 2.** (Continued)

| Data category | Information |
|---|---|
| Recruitment status | Recruiting |
| Primary outcome(s) | Patient-oriented Composite Endpoint (POCE) defined as the composite of all-cause death, any stroke, any myocardial infarction (MI)*, any repeat revascularization [Time Frame: 2 years follow-up] |
| Key secondary outcomes | 1) Device-oriented Composite Endpoint (DoCE) defined as the composite of cardiovascular death, target-vessel MI, clinically indicated repeat revascularization of the target lesion |
| | 2) Vessel-oriented Composite Endpoint (VoCE) defined as the composite of cardiovascular death, target vessel MI, repeat revascularization of the target vessel |
| | 3) PoCE at 1 year |
| | 4) All-individual components of PoCE |
| | 5) All individual components of DoCE |
| | 6) All individual components of VoCE |
| | 7) Definite and probable stent thrombosis according to ARCII definition (30) [28] |
| | 8) Investigator reported hospitalization for heart failure |
| | All outcomes will be reported at 1 and 2 years. |

## Study setting

Data will be collected from approximately 30 European cardiac centers in United Kingdom, Italy and Spain. A list of the study sites can be obtained at ClinicalTrials.gov (NCT04111770).

## Eligibility criteria

Patients with de novo lesion in an unprotected LMCA (ostial, shaft, or distal); or ostial LAD or ostial LCx, both compatible with one Medina class of LMCA disease; or ostial intermediate branch disease will be considered for enrolment. Subjects are considered for enrolment only after that PCI has been considered appropriate and feasible by the treating interventionalist. Enrolment criteria will be all-comers, with minimal inclusion and exclusion criteria (**Tables 3 and 4**). Eligible patients must have silent ischemia, stable angina, unstable angina, or non-ST-segment elevation myocardial infarction (NSTEMI). Patients with previous CABG can be considered for inclusion in case of failure of any of the grafts to one of the LMCA-bifurcation branches.

## Informed consent

Patients who are eligible for the study will be approached by the Investigator, or a qualified designee from the study research staff, and the study will be explained to them. If the patient is interested in participating, the study team will perform a more detailed screening of the patient and if they are still eligible, they will be given a patient information sheet to review. Potential subjects will be encouraged to discuss their possible participation with friends, family, and their family doctor. The contact details of the study team will be given to the patient in order to get in touch with them and arrange a visit (if they are still interested after reading and discussing the patient information in detail). Before any study specific procedures are performed, the patient will have time ask any questions and discuss any concerns regarding the study. Once all questions have been sufficiently asked and the patient is willing to proceed, the patient and the Investigator will sign and date the informed consent form (ICF).

It is anticipated that deferring 'ad hoc' revascularization to a second procedure after the diagnostic angiogram, to allow formal consent acquisition as described above, could represent

| | STUDY PERIOD | | | | | |
|---|---|---|---|---|---|---|
| | Enrolment | Allocation - PCI procedure | Post-allocation | | | Close-out |
| **TIMEPOINT** | | Within 24h after randomization or 72h for stable patients | Within 24h post-PCI or pre-discharge | 30d post-PCI (+7d) | 1y post-PCI (+30d) | *2y post-PCI (+30d)* |
| **TYPE OF CONTACT** | Visit | Visit | Visit | Phone | Visit or phone | Visit or phone |
| **ENROLMENT:** | | | | | | |
| **Eligibility screen** | X | | | | | |
| **Informed consent** | X | | | | | |
| **Vital signs** | X | | | | | |
| **Medical and Cardiac History** | X | | | | | |
| **Blood laboratory** | X | | | | | |
| **Cardiac Biomarkers** | X | | X | | | |
| **12-lead ECG** | X | | X | | X | X |
| **Left Ventricular Ejection Fraction & Valvular Disease Status** | X | | | | | |
| **Cardiac medications** | X | X | X | X | X | X |
| **Anginal status** | X | | | | | |
| **Enrollment & randomization** | X | | | | | |
| **Angiography** | X | X | | | | |
| **INTERVENTIONS:** | | | | | | |
| **Angiography** | X | X | | | | |
| **Intravascular Ultrasound** | | X | X | | X | |
| **ASSESSMENTS:** | | | | | | |
| *SYNTAX score* | | | | | | |
| *Serious adverse events reporting* | X | X | X | X | X | X |

**Fig 1. Schedule of enrolment, interventions, and assessments.** (PCI: percutaneous coronary intervention; ECG: electrocardiogram; IVUS: intravascular ultrasound).

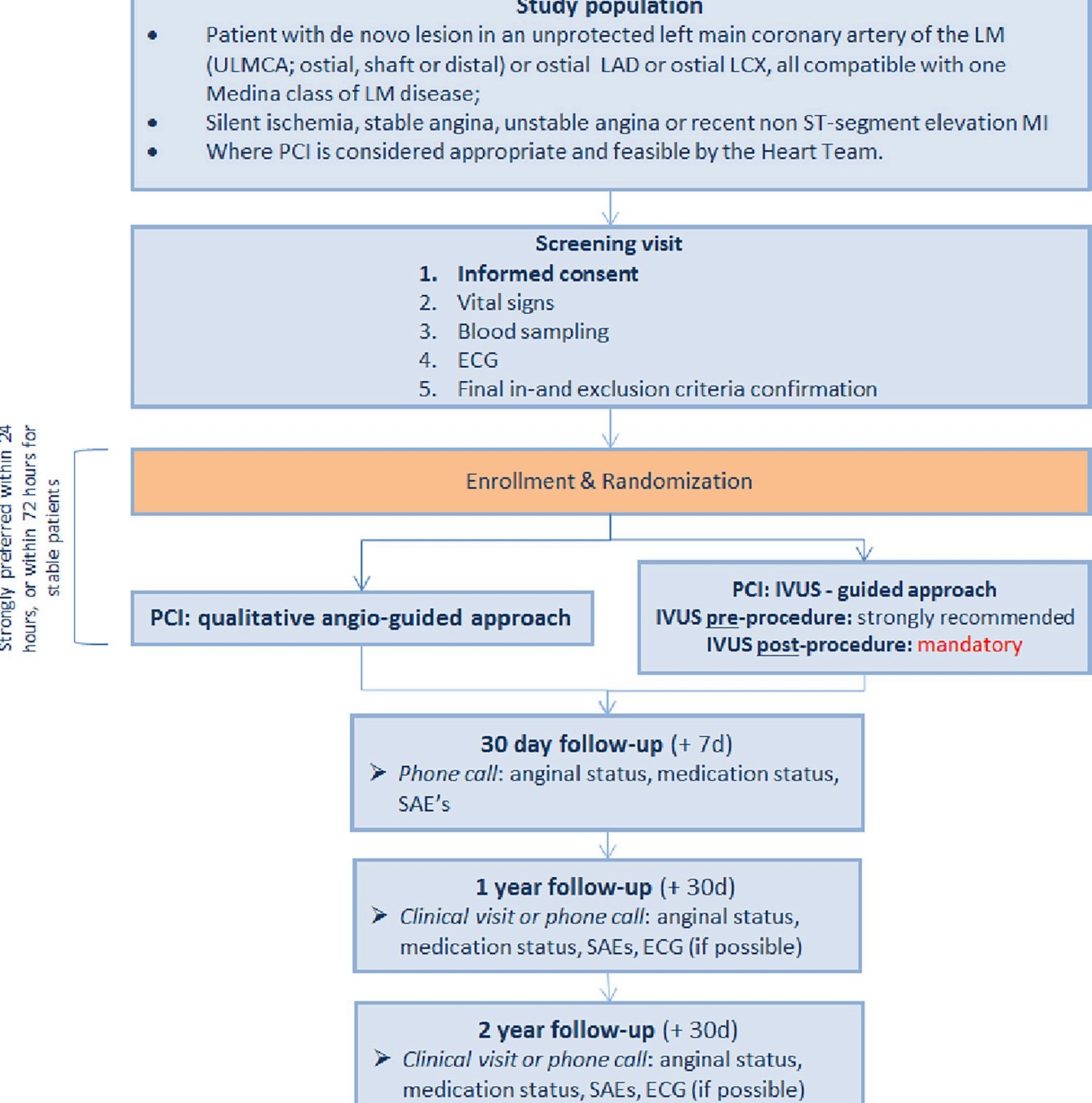

**Fig 2. Flow diagram showing patient inclusion, randomization and follow-up in the OPTIMAL trial.** (IVUS: intravascular ultrasound; LAD: left anterior descending; LCx: left circumflex; LM: left main; MI: myocardial infarction; PCI: percutaneous coronary intervention; SAE: serious adverse event; ULMCA: unprotected left main coronary artery).

**Table 3. Inclusion criteria.**

| |
|---|
| 1. The patient must be ≥18 years of age; |
| 2. De novo lesion in an unprotected LMCA (ULMCA; ostial, shaft or distal) OR ostial LAD or ostial LCx, both compatible with one Medina class of LMCA disease; or ostial intermediate branch disease. |
| 3. PCI is considered appropriate and feasible by the treating interventionalist; |
| 4. Silent ischemia, stable angina, unstable angina or non-ST-segment elevation MI (NSTEMI) |
| 5. Able to understand and provide informed consent and comply with all study procedures, including follow-up for at least 2 years. |
| Note: A patient with a prior CABG with no patent bypass on the LMCA can be included. |

a deviation from standard clinical routine. This is especially relevant during the COVID-19 pandemic when there is a clear interest in preventing prolonged hospitalizations unless strictly required. For this reason, in the case of a subject in need of 'ad hoc' revascularization, such as in NSTEMI, a staged informed consent procedure can be contemplated. More specifically, subjects that meet the eligibility criteria will be orally informed about the nature of the study by one of the investigators, witnessed at all times by independent catheterization laboratory personnel. The subject will provide oral informed consent, and both the investigator and the witness will sign the shortened informed consent before any study procedure takes place. Directly after the procedure, within a maximum of 24 hours, the subject will be fully informed by means of the complete patient informed consent, as already described.

## Ethics approval

The study adheres to the ethical principles of the Declaration of Helsinki, Good Clinical Practice (GCP) and to applicable European regulations, including ISO 14155:2020 and MED-DEV2.12. The trial starts locally at the study sites only after approval by the (I)ECs and other regulatory bodies as required per local regulations.

## Interventions

**Pre-PCI.** At baseline, medical/cardiac history and cardiac medications, vital signs (< 72 hours prior to PCI), electrocardiogram (ECG) (< 24 hours prior to PCI), anginal status (< 24

**Table 4. Exclusion criteria.**

| |
|---|
| 1. Patient is a woman who is pregnant or nursing; |
| 2. Female patient of childbearing potential, i.e., who are not surgically sterile or post-menopausal (defined as no menses for 2 years without an alternative cause); |
| 3. IVUS is strictly required for pre-PCI lesion severity assessment; |
| 4. ST-elevation MI, cardiogenic shock; |
| 5. Previous history of CABG with patent graft to the LAD and/or patent graft to the LCx; |
| 6. Prior PCI of the LMCA, ostial LAD or ostial LCx at any time prior to enrolment; |
| 7. Prior PCI of any other (i.e., non-LMCA, non-ostial-LAD and non-ostial-LCx) coronary artery lesions within 30 days prior to enrolment; |
| 8. Subjects unable to tolerate, obtain or comply with dual antiplatelet therapy for at least 1 year |
| 9. Known contraindication or hypersensitivity to everolimus, platinum-chromium, or to anticoagulants. |
| 10. Patients requiring additional surgery (cardiac or non-cardiac) within 3 months post-enrolment; |
| 11. Non-cardiac co-morbidities with a life expectancy less than 2 years; |
| 12. Currently participating in another trial that is not yet at its primary outcome. The patient is not allowed to participate in another investigational device or drug study for at least 12 months after enrolment. |

hours to PCI) and ejection fraction and valvular function (< 28 days prior to enrolment) will be recorded. Anginal status will be assessed using the Braunwald classification of unstable angina [29] and Canadian Cardiovascular Society classification of exertional angina [30]. Baseline evaluation will include pre-procedural routine laboratory tests according to local hospital practice. Serum creatinine and estimation of creatinine clearance (Cockcroft and Gault) are mandatory prior to procedure. Twelve-lead ECGs before and after the procedure, as well as at discharge, are also mandatory. Baseline cardiac biomarkers, including Troponins or Creatine-kinase-MB (CK-MB), are measured < 24 hours prior to PCI (< 72 hours allowed for stable patients). If elevated at baseline, an additional blood sample is recommended prior to PCI (when clinically possible) to determine if biomarkers are stable, decreasing, or increasing. All patients, unless already on chronic therapy with aspirin, will be preloaded with aspirin (300–325 mg) more than 2 hours prior to PCI. Pre-loading with a second antiplatelet agent (Clopidogrel 600 mg or Prasugrel 60 mg or Ticagrelor 180 mg) more than 2 hours prior to PCI is also mandatory unless patients are already on chronic therapy with either of these agents. For participants already receiving chronic ADP antagonist therapy, pre-loading should follow current guidelines and local practice. The choice of the second antiplatelet agent is left to the discretion of the Investigator. Unless contraindicated, periprocedural glycoprotein lIb/lIIa inhibitors will be given according to guidelines and at operator's discretion.

**At the time of PCI.** PCI on LMCA is to be scheduled within 24 hours after randomization, or within 72 hours for stable angina patients. Intervention on LMCA is indicated in cases of "visually-defined" angiographic diameter stenosis (DS) $\geq$ 70%, or 50% < DS < 70% in the presence of inducible ischemia at non-invasive test or at pressure wire study (FFR $\leq$ 0.80 or iFR < 0.90 or resting full-cycle ratio (RFR) < 0.90). Indication of PCI on LMCA can also be based on IVUS evidence of LMCA MLA $\leq$ 6 mm$^2$ at the time of diagnostic procedure (e.g., at the time of angiogram performed before Heart Team discussion and enrolment into the study).

The use of IVUS before PCI to assess lesion severity is not allowed before randomization and in patients randomized to the angiography-guidance arm of the study.

The selection of best mode for lesion preparation and the decision about use of any hemodynamic support, either planned or urgently required, are left to the operator's best judgment. Choice of vascular access (radial vs femoral) is also left to operator's discretion. In case of concomitant non-LMCA disease, this should be addressed with PCI before intervention on LMCA, unless intervention on LMCA should take priority because of very critical stenosis (e.g., DS > 90%) or LMCA-disease-related clinical instability. Revascularization to chronic total occlusion is advised after PCI to LMCA, as a staged procedure. Intervention on non-LMCA disease should be based on the same criteria considered for indication of PCI on LMCA. An MLA $\leq$ 4.0 mm$^2$ can also be used to guide intervention on non-LMCA lesions in patients allocated to the IVUS-guided arm.

Every staged procedure should be performed within 45 days from completion of the index procedure and in the presence of stability of symptoms between the first and the subsequent procedure(s). Ultimately, an independent Clinical Events Committee (CEC) determines if the second procedure is counted as a staged procedure (i.e., included in the index treatment) or as a revascularization (i.e. counted as an outcome) [31]. Even though the stent system is not under investigation in the OPTIMAL study, PCI either on LMCA as on non-LMCA disease is to be performed using either the Synergy or Synergy Megatron stent (Boston Scientific, Marlborough, Massachusetts, USA).

To ensure that the interventions are performed according to the guidelines in the various study sites, Steering Committee or delegates will be reviewing angiogram and IVUS images and provide feedback when required.

**Stenting technique in LMCA-bifurcation PCI.** Provisional cross-over stenting is to be considered as first-line strategy for LMCA bifurcation, with stenting from LMCA into LAD in cases of 1) small non-dominant LCx; 2) wide angle between LAD–LCx; 3) LCx ostium free of disease (defined as angiographic DS% < 50% and disease extending < 5 mm or as LCx ostium MLA > 4 mm$^2$ and plaque burden at the ostium < 50% [when pre-PCI IVUS is used in the IVUS-guided arm]). Provisional cross-over stenting from LMCA into LCx is to be considered when a large (> 3 mm diameter) LCx is diseased in the presence of LAD ostium free of disease (defined as angiography DS% < 50% and disease extending < 5 mm or as LAD ostium MLA > 4 mm$^2$ and plaque burden at the ostium < 50% [when pre-PCI IVUS is used in the IVUS-guided arm]).

Proximal and distal optimization techniques (POT & DOT) are mandatory both in IVUS-guided as in angiography-guided arms. Final kissing inflation (FKI) with non-compliant balloons is recommended in case of provisional stenting. If FKI is performed, then final POT is required (POT-FKI-POT).

Double stenting technique as bail-out after failed provisional stenting should be considered if side branch presents either 1) Angiographic DS% > 75%; 2) impaired TIMI flow (TIMI < 3); 3) ostial dissection; 4) FFR ≤ 0.80 or iFR < 0.90 or RFR < 0.90; or 5) Ostium MLA ≤ 4 mm$^2$, with plaque burden > 50% at post-stenting IVUS on side branch (in IVUS-guided arm, only). On the contrary, double stent technique should be considered as a first-line approach in cases of dominant or large caliber LCx (≥ 3 mm diameter) and with both LAD and LCx presenting 1) ostial angiographic DS% > 50% & disease extending > 5 mm; 2) MLA ≤ 4 mm$^2$ and plaque burden at the ostium > 50% at pre-PCI IVUS, if performed (in IVUS-guided arm, only). FKI with non-compliant balloons is mandatory in double stent technique and should be followed by final POT (POT-FKI-POT).

The choice of a particular distal bifurcation stent strategy is left to the operator's best judgment and may include T-stenting, T with a protrusion (TAP), crush, double kissing (DK)-crush or culotte stent techniques. A "V-stent" distal bifurcation stent approach and simultaneous kissing stents (SKS) are strongly discouraged.

**Angiography-guided optimization.** Criteria for optimal PCI of LMCA and non-LMCA lesions in the angio-guided arm are as follows: attainment of a final in-stent residual stenosis and edge stenosis < 30% as observed by quantitative coronary angiography (QCA), or < 20% by visual estimation if QCA is not available. Postdilations with non-compliant balloons is highly recommended for PCI both on LMCA as on non-LMCA lesions, to guarantee optimal stent expansion and apposition.

In the angiography-guided arm the use of IVUS is strongly discouraged. The operator is however allowed to use IVUS if an imaging technique is strictly required for patient's safety, i.e. in those cases showing an unclear angiographic result where there is a chance for a suboptimal result or a potential complication.

**IVUS-guided optimization.** When performed before stenting, IVUS will be used to guide appropriate stent sizing (based on distal reference mean diameter up-rounded (0.00–0.25 mm) to the nearest stent size), appropriate distal landing (by selecting proximal and distal references as site free from atheroma, or with plaque burden < 50%) and appropriate sizing of balloons for postdilation, for POT & DOT (selected balloon diameter according to mean external elastic lamina (EEL) diameter at the site of MLA, proximal and distal reference respectively).

Post-procedural IVUS is mandatory in IVUS-guided PCI arm. All the stented LMCA segments (both stents in the case of 2-stent strategy) must be interrogated by IVUS before completion of the case. An MSA > 8mm$^2$ must be targeted at the body of the LMCA, > 7mm$^2$ at the Polygon of Confluence (POC); > 6mm$^2$ at the LAD ostium, and > 5mm$^2$ at the LCx ostium (Kang's criteria) [19]. In case of provisional crossover technique in the IVUS-guided

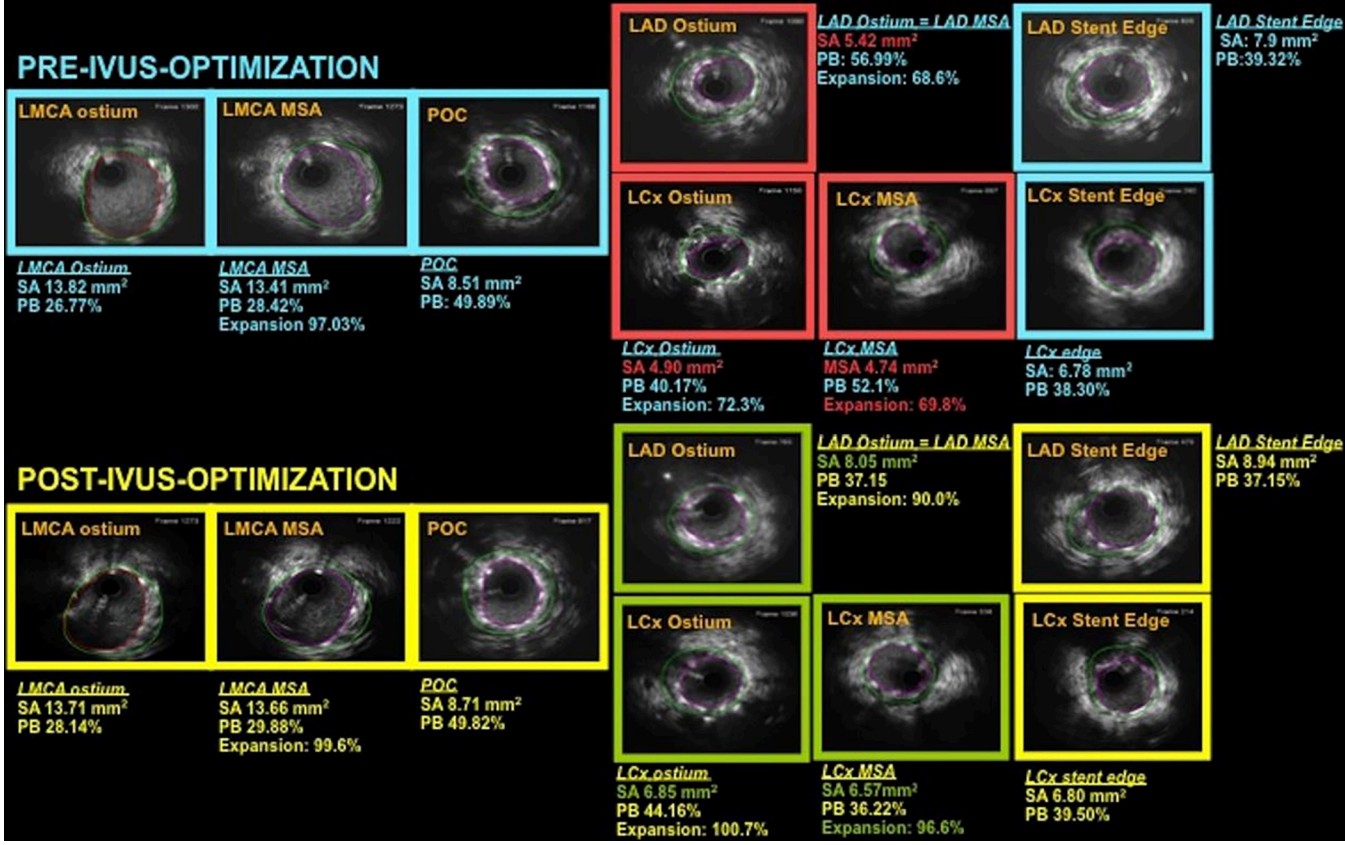

**Fig 3. Example of IVUS-guided stent optimization following the OPTIMAL-optimization criteria.** Before optimization, IVUS showed stent underexpansion at the ostium of the LAD (stent area 5.42 mm$^2$ [target > 6 mm$^2$]), at the ostium of the LCx (stent area 4.90 mm$^2$ [target > 5 mm$^2$]) and in the proximal LCx (minimum stent area 4.74 mm$^2$ [target > 5 mm$^2$] accounting for 69.8% of the distal reference area [target > 90%]) (Red boxes in the upper half of the figure). Further postdilation on the LAD and LCx was performed, completing with FKI and final POT. After optimization, IVUS confirmed achievement of stent expansion targets in all segments, including those who appeared initially suboptimally expanded (Green boxes in lowed half of the figure). (LAD: left anterior descending; LCx: left circumflex; LMCA: left main coronary artery; MSA: minimal stent area; PB: plaque burden; POC: polygon of convergence; SA: stent area).

group, IVUS is recommended also in the non–stented side branch (Fig 3). In general, whenever possible, postdilations with non-compliant balloons is highly recommended for PCI both on LMCA as on non-LMCA lesions, in order to guarantee appropriate stent expansion and apposition.

In patients randomized to IVUS-guidance and undergoing PCI to non-LMCA segments/vessels, IVUS post-stent optimization is highly recommended also in non-LMCA stented segments. In non-LMCA lesions optimization will be performed to achieve the following targets of appropriate expansion: MSA > 5.0 mm$^2$ and MSA > 80% of mean reference (for stenting on RCA or mid LAD/mid LCx away from LMCA [e.g., in cases of stent not overlapping the one deployed to LMCA bifurcation]), or MSA > 90% of distal reference (for stenting on mid LAD/LCx close to LMCA [e.g., in cases of stent overlapping the one deployed to LMCA bifurcation]).

Further IVUS-guided optimization on both LMCA and in non-LMCA should be considered to address: 1) significant stent edge dissection (flap > 60˚ and/or extending > 2 mm longitudinally); 2) geographical miss/incomplete plaque coverage (stent edge plaque burden > 50%); 3) struts malapposition (maximal malapposition distance > 0.4 mm and malapposition segment longitudinal extension > 1 mm [32]. IVUS assessment will be

performed with either Opticross HD system (Boston Scientific), or Refinity, Revolution or Eagle Eye Platinum systems (Philips Volcano).

**Post PCI.** Cardiac biomarkers shall be measured 6–24 hours post-PCI, and in patients with elevated post-PCI values (CK-MB > 5 upper limit normal [ULN], or cTroponin/high-sensitivity-cTroponin > 35 ULN) serial measurements must be taken until decline is noted. At the time of discharge, anginal status and any serious adverse event (SAE) will be recorded along with ECG and cardiac medications. From day 1 post-procedure, aspirin 75–100 mg daily will be prescribed indefinitely for all patients. Chronic therapy with second antiplatelet agent is mandatory for at least 6 months in stable patients and 1 year in acute coronary syndrome patients. Choice of second antiplatelet agent is at the investigator's discretion (Clopidogrel 75 mg/day, or Prasugrel 10 mg/day [or 5 mg/day if weight < 60 kg or age > 75 years], or Ticagrelor 90 mg twice a day). Patients with indication for oral anticoagulation should be treated per European Society of Cardiology (ESC) guidelines [21]. Optimal medical therapy (e.g., beta-blockers, angiotensin-converting enzyme [ACE] inhibitors) should be prescribed in accordance with current guidelines.

## Follow-up

The patient's clinical status will be assessed at discharge. Telephone follow-up will be made at 30 days (± 7 days), whilst hospital visits will occur at 1 year (± 30 days) and 2 years (± 30 days) post-procedure. Telephone follow-up might also take place at 1 and 2 years, if it is impossible to perform formal hospital visit follow-up. The follow-up will include assessment of angina status and recording of cardiac medications and any SAEs. A 12-lead ECG will also be performed (if possible). Information will be gathered with regard to any major adverse cardiac or cerebrovascular events (MACCE).

## Modifying allocated interventions

For participants randomized to QCA-guided approach, IVUS may be deemed necessary for clinical reasons. There may be complications during the PCI where IVUS is performed as a bailout or rescue procedure. Also, an IVUS has to be done if there are any unclear angiographic findings during PCI, such as a probable dissection, and probable incomplete stent coverage, or deployment. In these cases, IVUS is allowed to be performed and the patient will be considered a cross-over. The reason for performing IVUS should be clearly documented in the electronic data capture (EDC) system.

## Strategies to improve adherence to interventions

In this interventional trial, adherence to the assigned treatment strategy has a short window of opportunity. To ensure that patients randomized undergo PCI, a window of 24 hours for acute patients and 72 hours for stable patients has been defined between randomization and PCI. In the IVUS-guided arm, all patients undergo optimization based on protocol documentations. Post-procedural IVUS is mandatory. All the stented LMCA segments (both stents in the case of 2-stent strategy) must be interrogated by IVUS before completion of the case. In the non-LM segments/vessels, IVUS post-stent optimization is highly recommended. In the qualitative angio-guided PCI arm, post-procedural IVUS is strongly discouraged unless strictly medically required in angio-guided arm. All IVUS pullbacks are collected centrally and monitored by the Steering Committee. A detailed IVUS acquisition protocol is provided to the sites and comprehensive data on IVUS usage is entered in the eCRF, which is monitored centrally. The protocol also requires the use of Synergy family stents for which the grant-giver ensures

availability at the participating sites. Drug therapy in the context of PCI follows current European guidelines and is not investigational in this trial.

## Relevant concomitant care or interventions permitted or prohibited during the trial

**Staged procedures.** Given the complexity of the LMCA participants enrolled in this clinical trial, it is anticipated that a substantial proportion of participants may fall into the category of staged procedures. In the OPTIMAL trial, a staged procedure is defined as a planned intervention performed after the first catheterization. The procedure must be well documented in the EDC, must be performed within 45 days from the index PCI and the patient must be stable between the first and the subsequent procedures. In general, the decision to stage is based on patient factors (e.g., kidney function, contrast exposure, radiation exposure, patient fatigue), lesion complexity (e.g., calcifications), unexpected lengthy procedures, procedural complications, patient instability, or unsuccessful first attempt. Ultimately, an independent Clinical Events Committee (CEC) determines if the second procedure is counted as a staged procedure (i.e., included in the index treatment) or as a revascularization (i.e., counted as an endpoint). During a staged procedure, the same study assessments apply as during the baseline procedure. The staged procedures will not affect the original follow-up schedule.

**Hemodynamic support.** Hemodynamic support (e.g., intra-aortic balloon pump) during PCI for LMCA lesions is usually not required. Criteria for required hemodynamic support may include systemic hypotension, severe pulmonary hypertension, severely reduced ejection fraction, extreme anatomic complexity (e.g., severely calcified left main lesion with intended use of rotational atherectomy), and/or participant instability before or during the procedure. The decision regarding the use hemodynamic support and the type of support device is left to the operator's best judgment.

**Optimal PCI of other coronary lesions.** Participants should be revascularized optimally based on the Investigator's discretion. For all "borderline or intermediate non-left main lesions" (i.e., 40–70% diameter stenosis by angiographic visual estimate), it is strongly recommended to confirm the lesion significance before treatment using physiology (preferred, including FFR, iFR or DFR) or IVUS assessment (alternate). If they are not severe, PCI should not be performed. For all non-left main lesions, when the participants are allocated to the IVUS guidance arm, IVUS guidance pre-treatment and assessment post-treatment to optimize lumen dimensions is recommended.

The liberal use of additional guidewires to protect side branches during complex angioplasty is recommended as per the operator's best judgment. Lesion preparation using balloons, or any approved device is left to the operator's best judgment to be able to deliver the stent to the lesion and achieve full stent expansion. It is mandatory that only Synergy or Synergy Megatron DES is used for all non-left main coronary lesions which are stented. If the Synergy or Synergy Megatron DES is either unavailable or cannot be delivered to lesion site, it is recommended to follow standard of care at the site. Liberal use of short non-compliant post-dilation balloons ($\geq$18 atm) within the stent margins of all stents is recommended to optimize luminal results, unless IVUS otherwise shows optimal expansion and lumen dimensions.

Criteria for optimal PCI of LMCA and non-LM lesions are as follows: attainment of a final in-stent residual stenosis and edge stenosis of <30% as observed by QCA or <20% by visual estimation if QCA is not available. Specific lesion categories, such as chronic total occlusions, bifurcation disease, diffuse lesions, thrombus-containing lesions or heavily calcified lesions should be treated according to the operator's best judgment using acknowledged best PCI practices. In situations of diffuse disease or tandem lesions it is recommended to use single

long stents rather than either two shorter side-by-side or overlapping stents. A liberal staging strategy should be used, especially for participants with complex double vessel or triple vessel disease, or if high doses of radiation and/or contrast are used in the first procedure.

**Pharmacological therapy: Anticoagulants, antiplatelets.**  Unfractionated heparin or low molecular weight heparin is acceptable, according to proper dose guidelines and adjusted for renal insufficiency. Fondaparinux is not permitted as a procedural anticoagulant. Procedural anticoagulants should usually be discontinued at the end of the procedure, but in rare cases may be continued at low dose as per physician discretion (e.g., for participants with an indwelling intra-aortic balloon pump). The routine use of post-procedural low molecular weight heparin for prophylaxis of deep venous thrombosis is not permitted. Participants with indication for OAC should be treated as described in the current ESC guidelines, e.g., triple therapy up to 6 months (or 1 month in case of high bleeding risk), followed by dual therapy up to 12 months.

GP IIb/IIIa inhibitors are strongly discouraged in participants adequately pre-loaded with an ADP antagonist (clopidogrel, prasugrel, or ticagrelor). GP IIb/IIIa inhibitors may be used, however, in participants with large amounts of thrombus, or for thrombotic or ischemic complications arising during the procedure. Provisional GP IIb/IIIa inhibitors may not be used for "soft" indications such as lesion haziness or a small dissection as their use in these situations will increase bleeding complications, without clear benefit.

Chronic daily adenosine-diphosphate (ADP) antagonist therapy (e.g., clopidogrel, prasugrel, ticagrelor) is mandated for a minimum of 6 months in stable patients and 1 year in ACS patients after PCI. The choice of agent is left to the discretion of the Investigator, local standard of care and drug availability. ADP antagonists should not be discontinued after DES implantation unless absolutely necessary for major bleeding, major trauma, or major surgery necessitating discontinuation of antiplatelet therapy (e.g., intracranial surgery). Additionally, following the PCI procedure, all participants should continue on aspirin (minimum of 75 mg/day up to 162 mg/day or dose per standard hospital practice) indefinitely. Aspirin should not be discontinued for CABG or other reasons unless absolutely necessary.

**Provisions for post-trial care and trial-related injuries.**  Once the 2-year follow-up of the participants has finished, the participants will be followed as per local hospital standard of care. Continued provision of the intervention for participants is not applicable. In the event of trial-related injuries, ECRI maintains clinical trial insurance coverage for the participants, in accordance with the applicable laws and regulations of the country in which the procedure is performed.

## Outcomes

The primary outcome is the ARC PoCE at 2-years follow-up [28]. PoCE is defined as: all-cause death, any stroke, any MI (using the Society for Cardiovascular Angiography and Interventions [SCAI] definition for peri-procedural MI and the 4th universal definition for spontaneous MI [UDMI] for MIs after 48 hours) [33, 34], any repeat revascularization. The secondary outcomes include: 1) Device-oriented Composite Endpoint (DoCE) defined as the composite of: cardiovascular death, target-vessel MI, clinically-indicated repeat revascularization of the target lesion; 2) Vessel-oriented Composite Endpoint (VoCE) defined as the composite of: cardiovascular death, target vessel MI, repeat revascularization of the target vessel; 3) PoCE at 1 year; 4) All-individual components of PoCE; 5) All individual components of DoCE; 6) All individual components of VoCE; 7) Definite and probable stent thrombosis according to ARCII definition [28]; 8) Investigator reported hospitalization for heart failure. All outcomes will be reported at 1 and 2 years.

## Sample size

Based on previous literature, a 35% reduction of PoCE by using IVUS guidance in Left Main PCI is anticipated [23, 35, 36]. Based on data of the EXCEL trial, the expected 2-year PoCE in the IVUS-guided arm is assumed to be 17%, whereas PoCE in the angiography-guided arm is assumed to be 26% [4]. It is noteworthy that the OPTIMAL trial will not use the EXCEL MI definition, but it is expected that the combined SCAI/UDMI approach will yield comparable results [37]. The cross-over rate from angio-guidance to IVUS guidance is assumed to be 4%, and the cross-over rate from IVUS guidance to angio-guidance 3%. Taking these cross-over rates into account and using a Z (standard normal) test for the difference between two proportions and a two-sided type I error of 0.05, approximately 750 participants are needed to demonstrate superiority of an IVUS-guided approach versus an angiography-guided one with a statistical power of 80%. A validated on-line calculator was used [38]. A sample size of 2 x 399 participants allows for an attrition rate of 6%. The final sample size was set at 800.

## Recruitment

Potential participants will be identified by the attending clinician or a member of the clinical care team, who are part of the study research staff. This will mainly in the hospital setting prior to PCI procedure. There is no intention to use publicity through posters, leaflets, adverts or websites for recruitment of potential participants.

## Assignment of interventions: Sequence generation, allocation, blinding

Once the ICF is signed, the Investigator or designee registers the participant in the electronic data capturing system (EDC). The EDC assigns to the participant a unique identification number. After enrollment and after final eligibility is established, the participant is randomized at a 1:1 ratio to IVUS-guided PCI or qualitative angio-guided PCI using the randomization module of the EDC. The participants are randomized at a 1:1 ratio to IVUS-guided PCI or qualitative angio-guided PCI. Stratification by site is performed to ensure balance across potential local differences in treatment practices. If after informed consent, final eligibility cannot be established, the patient is considered a screening failure and is not randomized. All randomized patients are followed-up and included in the Full Analysis Set population.

It is strongly recommended that the PCI procedure is scheduled within 24 hours after informed consent, or within 72 hours for stable patients. If a consented patient is randomized and treated after 72 hours, reasons must be documented in the EDC. A consented patient that is not randomized within 4 weeks, will no longer be eligible and considered a screening failure.

The OPTIMAL is an open trial, where the patients and investigators are not blinded for the procedure. The clinical events committee will review the cases and adjudicate in a blinded manner, consistent with the prospective randomized open blinded endpoint (PROBE) approach.

## Data collection and management

The data recording tool for this study is a validated EDC system. All access to the EDC system is password-protected. Relevant study personnel seeking access to the EDC system follows a training before access is granted. The Investigator maintains an authorized signature log of appropriately qualified and trained site personnel to whom study duties have been delegated. All site personnel authorized to make entries and/or corrections on the EDC system are included on the authorized signature log. All personnel with access to the EDC system is supported by a Service Desk. The EDC contains a system-generated audit trail that captures any

changes made to a data field, including who made the change, why the change was made and the date and time it was made.

All data entry into the EDC system should be completed *within 5 business days* after the participant's visit/contact, or within 3 calendar days in case of SAE, to enable the monitor to review the participant's status throughout the study. The participant's data in the EDC system will be completed by Investigator or qualified designee and reviewed and signed off (e-signed) by the Investigator. Data entries made in the EDC system are supported by source documents maintained for all participants enrolled in this study at the site. EDC system completion guidelines are provided to the study site staff, before the first participant is enrolled at that site.

## Plans to promote participant retention and complete follow-up

**Missed visits.** Every effort will be made to ensure participants adhere to the follow-up. If the participant is unable to return for an onsite clinical visit or have a follow-up phone call, the Investigator or qualified designee must document the reason, and make reasonable effort to obtain the information from the participant otherwise.

**Lost to follow-up.** A participant is considered lost to follow-up when contact with the participant has been lost without completing the final assessment at the end of the 2-year follow-up period, and every attempt to contact has failed. At least 3 documented attempts must be made to contact the participant at each follow-up timepoint. Collection of any available data or vital status, either at the interventional center, at the referring hospital, with the general practitioner, or the municipal registries continues. Information regarding all attempts to contact the participant are documented and an end of study form must be completed. Participants who are lost to follow-up will not be replaced.

**Withdrawals.** Participants who explicitly disallow further clinical follow-up and data collection are considered "withdrawals". If a participant disallows further collection of study data, their data are evaluated until then. The investigator will make a reasonable effort to ascertain the reason(s), while fully respecting the participant's rights. If available, the reason is recorded, and an end-of-study form is completed. Participant-specific data on the basis of material obtained before withdrawal may be generated after withdrawal (e.g., image reading, analysis of blood sample); these data will also be retained and statistically analyzed in accordance with the statistical analysis plan.

**Data management.** All data entered into the EDC has source documentation available at the site. There is also a comprehensive and centralized file (Investigator Site File) of all study-related essential documentation, suitable for inspection at any time by representatives from sponsor and/or applicable regulatory authorities. Only authorized personnel, auditors and monitors have access to the study data. Clinical data management is performed in accordance with applicable Contract Research Organization's (CRO's) standards and data cleaning procedures (Cardialysis, B.V.). Additionally, each clinical site performs internal quality management of study conduct, data and biological specimen collection, documentation and completion. Quality control procedures are implemented beginning with the EDC and data QC checks that are run on the database. Any missing data or data anomalies are communicated to the site (s) for clarification/resolution. In addition, monitoring visits and possibly audits and inspections ensure oversight of the full quality control process.

**Confidentiality.** All information identifying the patients will be kept confidential and will not be supplied to the public. Names of the participants will not be made available to the sponsor. Only the participant's identifying number will be recorded in the EDC, and in case the participant's name appears on any document, including angiograms, IVUS, or ECG, it must be obliterated before a copy of it is made available to the sponsor. Additionally, all images sent

to Steering Committee or delegates for feedback are pseudo-anonymized and include only the patient's date of birth. Any other information that can identify the patient is removed. All study findings are stored in computers according to the local data protection laws. In case the study results are published, the participants' identities will remain confidential. However, the investigator is allowed to maintain a list to enable participants' identifications. For the purpose of the study, it is necessary to be able to trace data to the individual participants for 15 years, and thus, a participant identification code list will be available to link the data to the participants. The code will not include the participants' initials, and/or birth date. The key to the code will be safeguarded by the Investigator. Personal data handling will follow the EU General Data Protection Regulation and the national regulations.

## Statistical methods

Analysis of clinical primary and secondary outcomes is performed. The statistical analysis of the primary outcome is confirmatory. The statistical analysis of the secondary outcomes is exploratory. Two types of time-to-event, Kaplan-Meier (KM), analyses are performed. When the KM-curves for the IVUS-guidance arm and the Angio-Guidance arm are compared the regular KM-analysis is performed to compare the survival distributions of the two arms, using the log-rank test. This method is non-parametric. Additionally, the KM-estimates for the two arms at a single fixed time point τ (1 month, 12 months, 24 months) are compared, thus disregarding the shape of the survival curves up to time point τ. In this analysis the Com-Nougue approach will be applied. In the Com-Nougue approach the survival proportion at time point τ is calculated for both arms. A 95% Confidence interval and two-sided p-value will be calculated for the Risk Difference, being the difference in these survival proportions, are constructed using the Greenwood Standard Error of the survival proportions, as estimated by KM-analysis. In the calculation, the complement of the survival proportions is used, measuring the incidence of the reported event. Continuous variables will be presented using mean± standard deviation, median with interquartile ranges and minimum and maximum. In case Hazard Ratios are reported, the Cox proportional hazards model will be applied. Stratified analyses according to acute coronary syndrome, sex, SYNTAX Score I, distal LM bifurcation and the presence or absence of diabetes will also be carried out. Further details of the statistical methods and analysis are described in a statistical analysis plan (SAP).

**Methods in analysis to handle protocol non-adherence.** Primary analysis will be performed on in the Modified Intention-To-Treat population. For both the Per-Protocol population and As-Treated population only the Primary outcome, and its individual components, will be tabulated (no formal testing will be applied). The modified intention-to-treat analysis population will include all randomized participants who underwent the PCI procedure. The per-protocol population will include all randomized participants in whom PCI was performed according to their randomized arm, thus excluding crossovers, and participants of which the LMCA was not treated with PCI. The "as treated" population will include all randomized participants who underwent the PCI procedure, participants will be reported according to the actual treatment received.

**Plans to give access to the full protocol, participant-level data and statistical code.** The full protocol will be submitted together with the main publication, as required by the target journal. The participant-level dataset and statistical code will not be made publicly accessible but remain with the Sponsor.

## Oversight and monitoring

The OPTIMAL trial is an investigator-initiated study sponsored by the European Cardiovascular Research Institute (ECRI, Rotterdam, The Netherlands), and funded by Boston Scientific

(Marlborough, Massachusetts, USA) and Philips Volcano (Zaventem, Belgium). The Steering Committee designed the trial and oversees the medical, scientific, and operational conduct of the study, together with ECRI representatives and delegates. The Steering Committee is responsible for the overall design, conduct, and supervision of the study, including the development of any protocol amendments. The Steering Committee also reviews the progress of the study at regular intervals to ensure participant safety and study integrity. All trial operations including project management, data management, safety reporting, and statistical analyses are performed at Cardialysis (Rotterdam, The Netherlands). Site management and monitoring are performed as a joint effort between Cardialysis and the Principal Investigators. The composition of the Steering Committee and additional details are described in the Steering Committee charter.

Given that the study utilizes CE-mark devices widely used in Europe, including both the Synergy family stents (Boston Scientific) and the IVUS catheters (Philips Volcano), it was decided not to constitute a data and safety monitoring board (DSMB) for the study. In the OPTIMAL trial, the Steering Committee will perform the continuous monitoring of the conduct of the study. The Sponsor is responsible for the continuous assessment of the risk-benefit analysis throughout the study to ensure patient safety and study integrity.

The study may be temporarily suspended or prematurely terminated if there is sufficient reasonable cause per assessments by the Sponsor and in consultation with the Steering Committee. Written notification, documenting the reason for study suspension or termination, will be provided by the suspending or terminating party to Investigator, funding parties and regulatory authorities. If the study is prematurely terminated or suspended, the Investigator will promptly inform the (I)EC and will provide the reason(s) for the termination or suspension. Study participants will be contacted, as applicable, and informed of changes to study visit schedule. The study might resume once concerns are addressed and satisfy the suspending party. The sponsor will also notify the relevant regulatory authorities within 15 days, including the reasons for the premature termination. Within 1 year after the end of the study, the Investigator/sponsor will submit a final study report with the results of the study, or any publications/abstracts of the study, to the relevant regulatory authorities.

## Adverse event reporting and harms

An adverse event is defined as any untoward medical occurrence, disease, injury, clinical signs, or abnormal laboratory findings in participants. A serious adverse event (SAE) is an event that led to death, or serious deterioration in the health of a participant that resulted in a life-threatening illness or injury, permanent impairment, required hospitalization, or resulted in medical or surgical intervention to prevent permanent impairment. The Investigators will monitor the occurrence of adverse events for each participant throughout the course of the study. If an event fulfils the criteria for SAE, then this shall be reported in the EDC system without undue delay and within 3 calendar days of the site study staff's awareness, including the Investigators' judgment regarding causal relationship of the event to the trial procedure and/or the study device. Event-supporting source documents will be requested by the sponsor for the purpose of clinical event adjudication and safety reporting. All SAEs will be followed until the event has been resolved (with or without sequelae). Primary outcomes will be collected as SAEs and presented in periodic reports; however, they will be excluded from expedited reporting. Sponsor is responsible for the classification of SAEs and ongoing safety evaluation of the clinical investigation and shall review the Investigator's assessment of all SAEs and device deficiencies, determine and document in writing the sponsor's determination of seriousness and relationship to the investigational device.

If the Investigator observes device malfunctions that led or might have led to a death or serious deterioration in health of a participant, user or other person, or has complaints with regard to defects in the medical devices, the Investigator shall, within 24 hours of such observation, report such device malfunction or complaint to the device manufacturer, with a copy of the report to the Sponsor. The Sponsor is responsible for taking necessary actions in response to a device malfunction to protect the safety of the trial participants. The device manufacturer is responsible for handling all complaints and reported device malfunctions in respect of the quality of the medical device, including any measures deemed necessary, such as incident reporting to competent authorities and recalls. For safety reporting purposes, the list of anticipated adverse device effects is included in the protocol and serves as Reference Safety Information. ECRI sends a cumulative line listing to all principal investigators, relevant site staff and the central Ethics Committees every 6 months starting from the first patient in (FPI). Sites are requested to forward to local Ethics Committees. SAE processing, distribution and reporting are detailed in the safety reporting plan.

## Frequency and plans for auditing trial conduct

To ensure compliance with guidelines and regulations, a member of the CRO's quality assurance unit may arrange to conduct audits to assess the performance of the study at the study sites and of the study documents originating there. In addition, inspections by regulatory health authority representatives and (Independent) Ethics Committee(s) ([I]EC[s]), which are independent from investigators and the sponsor, are possible. Audits and inspections may occur at any time during or after completion of the study.

## Trial status

At the time of manuscript submission, the most recent protocol version is version 2.0, which was approved and signed on the 3rd of May 2021. The trial is in the recruiting phase. The first participant was enrolled on the 19th of July 2020. The overall time to enroll the total of 800 patients has been estimated of 2 years. The total study duration from the "first patient in" to "last patient out" is expected to be approximately 4 years.

## Discussion

### Protocol amendments

Any amendments to the study protocol that seem to be appropriate as the study progresses will be communicated to the Investigator by the Sponsor. All amendments will undergo the same review and approval process as the original protocol. All substantial and non-substantial amendments are handled according to local regulations and may be submitted for notification or approval to local authorities. Changes may be implemented after the protocol amendment has been approved by the IEC, unless immediate implementation of the change is necessary for participant safety.

### Dissemination plans

The Steering Committee and Investigators are committed to the publication and widespread dissemination of the results of the study. Data from this study will not be withheld regardless of the findings. All public presentations and manuscript generation and submissions will be led under the auspices of the Steering Committee. However, this study represents a joint effort between Investigators, ECRI and collaborators, and as such, the parties have agreed that the

recommendation of any party concerning manuscripts or text shall be taken into consideration in the preparation of final scientific documents for publication or presentation.

## Clinical perspectives

The OPTIMAL study will be the first and largest ever conducted clinical trial randomizing patients undergoing PCI to LMCA to IVUS-guidance versus angiography-guidance. Despite the widespread perception that IVUS-guidance can improve clinical outcomes, the evidence still comes from observational series or sub-analysis of other studies designed to address a different question. Moreover, the clinical benefit of IVUS-guidance has mostly been in terms of all-cause death rather than in terms of repeated revascularization or myocardial infarction. The latter remains somewhat elusive and difficult to explain fully. This is well reflected by the limited diffusion of IVUS in practice, especially in non-tertiary centers. Moreover, the current European and American Guidelines are still providing a IIa B recommendation to IVUS for both evaluation of LMCA disease and guidance of stent-optimization [22, 39]. The Level B of evidence calls in favor of a clinical trial, ad-hoc designed to address this matter.

For this reason, the OPTIMAL study has the potential to clarify once and for all the true role of IVUS- guidance on LMCA PCI. If the study will succeed in proving the superiority of IVUS-guidance in terms of long term POCE rate, it will be reasonable to expect a change in the grade of recommendation in the Guidelines, and a dramatic uptake of IVUS adoption in clinical practice.

## Supporting information

**S1 Appendix. SPIRIT (Standard Protocol Items: Recommendations for Interventional Trials) checklist completed for the OPTIMAL trial study protocol.** The SPIRIT 2013 Statement recommends the minimum content a clinical trial protocol should have. These evidence-based recommendations are widely considered as an international standard for study protocols. (DOCX)

**S2 Appendix.**
(PDF)

**S1 Protocol.**
(PDF)

**S1 File.**
(PDF)

## Author Contributions

**Conceptualization:** Giovanni Luigi De Maria, Luca Testa, Jose M. de la Torre Hernandez, Dimitrios Terentes-Printzios, Maria Emfietzoglou, Roberto Scarsini, Francesco Bedogni, Ernest Spitzer, Adrian Banning.

**Investigation:** Giovanni Luigi De Maria, Luca Testa, Jose M. de la Torre Hernandez, Dimitrios Terentes-Printzios, Maria Emfietzoglou, Roberto Scarsini, Francesco Bedogni, Adrian Banning.

**Methodology:** Giovanni Luigi De Maria, Jose M. de la Torre Hernandez, Dimitrios Terentes-Printzios, Maria Emfietzoglou, Roberto Scarsini, Ernest Spitzer, Adrian Banning.

**Project administration:** Ernest Spitzer.

**Supervision:** Giovanni Luigi De Maria, Jose M. de la Torre Hernandez, Ernest Spitzer, Adrian Banning.

**Writing – original draft:** Giovanni Luigi De Maria, Luca Testa, Jose M. de la Torre Hernandez, Dimitrios Terentes-Printzios, Maria Emfietzoglou, Roberto Scarsini, Francesco Bedogni, Ernest Spitzer, Adrian Banning.

**Writing – review & editing:** Giovanni Luigi De Maria, Luca Testa, Jose M. de la Torre Hernandez, Dimitrios Terentes-Printzios, Maria Emfietzoglou, Roberto Scarsini, Francesco Bedogni, Ernest Spitzer, Adrian Banning.

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
