## [Decision Letter · Decision Letter 0]

15 Oct 2021

PONE-D-21-13023

A Multi-Center, International, Randomized, 2-year, Parallel-group Study to Assess the Superiority of IVUS-Guided PCI Versus Qualitative Angio-Guided PCI in Unprotected Left Main Coronary Artery (LMCA) Disease: Study Protocol for OPTIMAL Trial

PLOS ONE

Dear Dr. Banning,

Thank you for submitting your manuscript to PLOS ONE. After careful consideration, we feel that it has merit but does not fully meet PLOS ONE’s publication criteria as it currently stands. Therefore, we invite you to submit a revised version of the manuscript that addresses the points raised during the review process.

We look forward to receiving your revised manuscript.

Kind regards,

Giuseppe Gargiulo, MD, PhD

Academic Editor

PLOS ONE

Additional Editor Comments (if provided):

This is the design paper of the OPTIMAL randomized trial, a large multicentre international investigator-initiated trial which is well designed and addressing a clinically relevant clinical question performed by a well known group of experts in the field. Methods described are appropriate, the trial was properly registered on clinicaltrials.gov, supported by the well known ECRI, including independent CRO (Cardialysis) with monitoring, independent CEC for adjudication of clinical events based on standard definitions. Sample size is properly described. Funding are properly disclosed. The authors also provide as appendix the full protocol approved.

The authors should be commended for running such an important trial with so rigorous methodology.

I have few comments to be considered in addition to those by the reviewers:

Please include as appendix if available the statistical analysis plan.

Please clarify which is the primary MI definition (SCAI or 4th UDMI?).

Please clarify in the text which software was used for sample size calculation based on the assumptions reported.

Please include more details on randomization (are there any blocks? How stratification is managed? Etc.).

Please update dates and number of patients enrolled.

I would suggest to include a small final section for a brief discussion and expected results (what is expected? How the study can advance the field? Are there similar ongoing trials? Etc.)

Journal Requirements:

3. Thank you for stating the following financial disclosure: "Yes. 

This is an Investigator-initiated study under the umbrella of the ECRI. Research grants have been provided by Boston Scientific and Philips Volcano to fund this study. ECRI maintains clinical trial insurance coverage for study participants in the event of trial-related injuries, if applicable and in accordance with the applicable laws and regulations of the country in which the study is performed. Also, according to the applicable regulatory requirement(s), ECRI provides insurance or indemnity (legal and financial coverage) to the Investigator/the Institution against claims arising from the trial, except for claims that arise from malpractice and/or negligence. "

4. Thank you for stating the following in the Competing Interests section: "Dr De Maria reports speaker fees from Miracor Medical SA. Dr Testa reports fees as medical proctor for Boston Scientific, Meril, Concept Medical, Abbott, Philips and advisory board member and/or speaker fees and/or institutional research grant from Boston Scientific, Philips, Abbott, Medtronic, Terumo, Concept Medical. Dr de la Torre Hernandez reports receipt of grants/research supports from Abbott Medical, Biosensors, Bristol Myers Squibb, Amgen and receipt of honoraria or consultation fees from Boston Scientific, Medtronic, Biotronik, Astra Zeneca, Daiichi-Sankyo. Dr Bedogni reports fees as medical proctor for BSCI, Meril, Medtronic, Terumo and advisory board member and/or speaker fees and/or institutional research grant from Boston Scientific, Philips, Abbott, Medtronic, Terumo, Concept Medical. Prof Banning reports institutional grant for fellowship form Boston and speaker fees Boston, Phillips and Miracor Medical SA. Dr Scarsini, Dr Terentes-Printzios, Dr Emfietzoglou, and Dr Spitzer have no conflict of interest to declare."

We note that you received funding from a commercial source: Boston Scientific Corporation (US)and Philips.

5. Please amend the manuscript submission data (via Edit Submission) to include author Jose M de la Torre Hernandez MD, PhD.

7. We note you have included a table to which you do not refer in the text of your manuscript. Please ensure that you refer to Table 1 & 5 in your text; if accepted, production will need this reference to link the reader to the Table.

Reviewers' comments:

Reviewer's Responses to Questions

**Comments to the Author**

1. Does the manuscript provide a valid rationale for the proposed study, with clearly identified and justified research questions?

Reviewer #1: Yes

Reviewer #2: Yes

Reviewer #3: Yes

2. Is the protocol technically sound and planned in a manner that will lead to a meaningful outcome and allow testing the stated hypotheses?

Reviewer #1: Yes

Reviewer #2: Yes

Reviewer #3: Yes

3. Is the methodology feasible and described in sufficient detail to allow the work to be replicable?

Reviewer #1: Yes

Reviewer #2: Yes

Reviewer #3: Yes

4. Have the authors described where all data underlying the findings will be made available when the study is complete?

Reviewer #1: No

Reviewer #2: Yes

Reviewer #3: Yes

5. Is the manuscript presented in an intelligible fashion and written in standard English?

Reviewer #1: Yes

Reviewer #2: Yes

Reviewer #3: Yes

6. Review Comments to the Author

You may also provide optional suggestions and comments to authors that they might find helpful in planning their study.

Reviewer #1: Authors described a study protocol paper for the OPTIMAL trial. The OPTIMAL trial is a multi-center, international, randomized study to assess the superiority of IVUS-guided PCI versus qualitative angio-guided PCI in unprotected left main coronary artery disease. The manuscript is very well written. I have only one comment for authors.

Comment 1. In sample size, authors described that “The cross-over rate from angio-guidance to IVUS guidance is assumed to be 4%, and the cross-over rate from IVUS guidance to angio-guidance 3%.” I understand that the cross-over from angio-guidance to IVUS guidance would occur in some cases because of safety. However, I cannot understand why the cross-over from IVUS-guidance to angio-guidance would occur. Authors should explain why the cross-over from IVUS-guidance to angio-guidance occur in this study.

Reviewer #2: A Multi-Center, International, Randomized, 2-year, Parallel-group Study to Assess the Superiority of IVUS-Guided PCI Versus Qualitative Angio-Guided PCI in Unprotected Left Main Coronary Artery (LMCA) Disease: Study Protocol for OPTIMAL Trial

Summary

This is a methods paper of a recruiting open-label randomised controlled trial investigating the use of intravascular ultrasound (IVUS) to guide left main stem (LMS) percutaneous coronary intervention (PCI) versus using angiography alone. The rationale and hypothesis are clearly stated, the research question is an important one, is novel and if completed (as one would expect) the results are likely to be highly impactful. There is now evidence of improved outcomes (in terms of reduced target vessel failure) if IVUS is used for PCI compared to angiographic guidance alone. Evidence specifically in left main stem PCI is limited to observational studies, in which there appears to be a benefit of the use of IVUS in improving long-term outcomes. Uptake in the use of IVUS in these cases remains limited though, even in major trials such as EXCEL1 and NOBLE2 where it was recommended. This is a much-anticipated trial which is likely to be practice changing.

Inclusion and exclusion criteria are clearly stated. The study intervention is use of a diagnostic device (intravascular ultrasound (IVUS)) to guide PCI, which would not be a therapeutic intervention if not acted upon. Importantly, the authors clearly state the (relatively conservative) minimal stented areas which should be aimed for in the IVUS guided group which would allow replication. Outcomes (both primary and secondary) are clearly defined, as are planned statistical methods. The authors have included a completed SPIRIT checklist and have addressed all relevant aspects of it.

Overall, this is a comprehensive methods paper for an important and what appears to be rigorous, open-label randomised controlled trial comparing the use of IVUS to guide PCI. Whilst importance/impact may not be the journal’s priority, it is worth saying that this trial is likely to be important to the practice of coronary intervention. More specifically related to the Journal’s publication criteria, this is a detailed and rigorous trial protocol. It complies with/satisfies all of PLOS ONE’s submission guidelines and recommended considerations for study protocols.

I just have one specific comment/question to the authors:

1) The authors state that they will assess angina status prior to PCI and at follow-up, but do not specify how they intend to do this. I would suggest that this is specified in the manuscript.

References:

1. Stone GW, Sabik JF, Serruys PW, et al. Everolimus-Eluting Stents or Bypass Surgery for Left Main Coronary Artery Disease. N Engl J Med. 2016;375(23):2223-2235.

2. Mäkikallio T, Holm NR, Lindsay M, et al. Percutaneous coronary angioplasty versus coronary artery bypass grafting in treatment of unprotected left main stenosis (NOBLE): a prospective, randomised, open-label, non-inferiority trial. Lancet. 2016;388(10061):2743-2752.

Reviewer #3: The article proposes that IVUS-Guided PCI is superior to qualitative angio-Guided PCI in unprotected

Left Main Coronary Artery disease in terms of reducing MACE during 2 years follow-up. To demonstrate this, they will include 800 patients with unprotected left main coronary artery disease in a randomized multicenter, international study. Patients will be randomized to IVUS-guided PCI or angio guided PCI.

Although one might think that it is unethical to randomize patients to undergo LMCA revascularization guided by IVUS or by angiography at present time, the fact is that clinical practice guidelines give IVUS a IIB recommendation in this setting, on the basis that there are few short randomized single-center series evaluating this issue and that most of the information has been extracted from meta-analysis of non-randomized studies. A large study including 1670 patients with left-main lesions treated with DES, (propensity score-matched analyses) showed that IVUS guidance was associated with reduced MACE within 3 years. However, MACE reduction was driven by all-cause mortality without a reduction in MI or TLR, thus leaving open the question regarding the mechanism of the observed survival benefit. Similarly, in the observational MAIN-COMPARE study, a trend for lower mortality was demonstrated, yet again without a difference in MI or TLR. In these series and meta-analysis, a repetitive reduction in mortality has been observed without a reduction of infarcts or TLR, which suggests confounding factors, since there is no mechanistic explanation for this decrease in mortality.

Moreover, in daily clinical practice IVUS after LMCA PCI is not performed in a great percentage of procedures. Even in studies like Noble, where IVUS was strongly recommended, it was performed only in 70% of procedures. Thus, it is not uncommon that IVUS is not performed for guiding LMCA PCI, probably related to a class IIB recommendation, which perhaps is of not enough strength for a significant number of operators.

For these reasons, this reviewer considers it appropriate to carry out this randomized study, since it will definitely demonstrate without doubts whether guiding LMCA PCI by IVUS reduces MACE or not in follow-up.

Major comments

Regarding Angiography-Guided Optimization, line 319. I find authors make much more emphasis in obtaining a good immediate result in IVUS guided patients than in angio guided patients. The criteria for IVUS guided optimization are quite long. However, there are only 3 lines explaining the criteria for optimal PCI of LMCA in the angio-guided. This protocol difference could account for better results in the IVUS guided group.

Similarly, there are no instructions for LMCA PCI when it doesn’t involve bifurcation. POT is mandatory in bifurcation technique. But there is no recommendation of postdilation in the rest of LMCA lesions. In order to achieve the best LMCA PCI results not only in the IVUS group (which will be postdilated if needed after IVUS final interrogation) but also in the angio group, it should be mandatory to postdilate with non-compliant balloons after stent implant (to avoid stent under-expansion or malaposition which could remain unnoticed in the angio group provided IVUS won’t be done in this group)

Line 250: All patients, including those already on chronic therapy with aspirin, will be preloaded with aspirin (300-325 mg) more than 2 hours prior to PCI. Pre-loading is also mandatory for all patients with a second antiplatelet agent (Clopidogrel 600 mg or Prasugrel 60 mg or Ticagrelor 180 mg) more than 2 hours prior to PCI (including patients already on chronic therapy with either of these agents).

• In the opinion of this reviewer this is not routine clinical practice and has no clinical justification. Moreover, it can impose a risk of bleeding in a patient if he/she receives a high loading dose of aspirin/ticagrelor/prasugrel while being chronically on this treatment

According to table 2 in page 6: Silent ischemia, stable angina, unstable angina or recent non-ST-segment elevation MI are inclusion criteria. However, ongoing myocardial infarction or recent myocardial infarction with rising cardiac biomarker levels are exclusion criteria.

• This two sentences may be confusing. A patient with recent non-ST-segment elevation MI may have rising cardiac biomarker levels.

Regarding the same aspect, in table 4 page 16 the sentence changes a bit: Ongoing myocardial infarction or recent MI with still evidence of rising cardiac biomarker levels.

• Please modify this sentence in table 2 in order it is clearly understood that a patient with recent non-ST-segment elevation MI can be included only if cardiac markers are already decreasing.

In table 2 page 7, according to point 5, this is an exclusion criteria: “Previous history of CABG with patent graft to the LAD and/or patent graft to the LCX”

However, in table 4 page 16, according to point 5 the exclusion criteria is: “Previous history of CABG with patent grafts to both LAD and LCx.”

• Please clarify this aspect, ¿can a patient be included if he/she has patent one graft to the LAD/LCX provided the other is occluded? According to lines 218-220 and according to table 4 it is possible, although according to table 2 page 7 it isn’t.

Regarding the stenting technique in LMCA-bifurcation PCI:

Page 19. Line 304: Double stenting technique as bail-out after failed provisional stenting should be considered if side branch presents either 1) Angiographic DS% > 75%; 2) impaired TIMI flow (TIMI < 3); 3) ostial dissection; 4) FFR ≤ 0.80 or iFR < 0.90 or RFR < 0.90; or 5) Ostium MLA ≤ 4 mm2, with plaque burden > 50% at post-stenting IVUS on side branch (in IVUS-guided arm, only)

• Does this mean that IVUS cannot be used in the angio-guided arm to decide if it is necessary double stent technique as bail-out? This decision must be taken only based on angiography or functional assessment though iFR, RFR or FFR in the angio-guided arm?

• If it is like this, it should be stated that use of IVUS for this purpose in the angio-guided arm is not allowed.

Minor comments

There is a little change between DoCE and VoCE definitions

1) Device-oriented Composite Endpoint (DoCE) is defined as the composite of cardiovascular death, target-vessel MI, clinically indicated repeat revascularization of the target lesion

2) Vessel-oriented Composite Endpoint (VoCE) is defined as the composite of cardiovascular death, target vessel MI, repeat revascularization of the target vessel

• The words “clinically indicated” may be confusing. What is important is if there was a repeat revascularization of the target lesion, not if it was clinically indicated. That words may make investigators think that if the repeat revascularization was not clinically indicated it should not be recorded.

7. PLOS authors have the option to publish the peer review history of their article (what does this mean?). If published, this will include your full peer review and any attached files.

Reviewer #1: No

Reviewer #2: No

Reviewer #3: No

---

## [Author Response · Author response to Decision Letter 0]

29 Oct 2021

Academic Editor’s comments: 

This is the design paper of the OPTIMAL randomized trial, a large multicentre international investigator-initiated trial which is well designed and addressing a clinically relevant clinical question performed by a well known group of experts in the field. Methods described are appropriate, the trial was properly registered on clinicaltrials.gov, supported by the well known ECRI, including independent CRO (Cardialysis) with monitoring, independent CEC for adjudication of clinical events based on standard definitions. Sample size is properly described. Funding are properly disclosed. The authors also provide as appendix the full protocol approved. The authors should be commended for running such an important trial with so rigorous methodology.

We thank the Editor for the positive feedback provided to the design and rationale of the study and for seeing, as we do, merit in running this clinical trial. 

1. Please include as appendix if available the statistical analysis plan.

At the moment of this submission, the statistical analysis plan was not finalized and thus is not part of the submission. This is in compliance with applicable regulations, which specify that the statistical analysis plan should be finalized before breaking the blind, which is planned to occur few years ahead.

2. Please clarify which is the primary MI definition (SCAI or 4th UDMI?).

Thank you for your comment. We apologize if this point was not clear in our original submission. For peri-procedural MI we will use the Society for Cardiovascular Angiography and Interventions (SCAI) definition (Moussa ID et al. J Am Coll Cardiol. 2013;62(17):1563-1570), whereas for sponteaneous (>48 hours) MI, we will use the 4th Universal Definition for spontaneous MI (UDMI) (Thygesen et al. J Am Coll Cardiol. 2018;72(18):2231-2264). This strategy is considered the “Option A” for adjudication of myocardial infarction occurring within 48h of revascularization as elucidated in the Critical Appraisal of Contemporary Clinical Endpoint Definitions in Coronary Intervention Trials: A Guidance Document (Spitzer et al. JACC: Cardiovascular Interventions. 2019; 12(9): 805-819). Please see page 31, lines 595-597 in the “Revised Manuscript with Tracked Changes”. 

3. Please clarify in the text which software was used for sample size calculation based on the assumptions reported.

We thank the Editor for this request of clarification which has offered us the opportunity to correct small typos. The current text reads as follows below:

Based on previous literature, a 35% reduction of PoCE by using IVUS guidance in Left Main PCI is anticipated [23, 35, 36]. Based on data of the EXCEL trial, the expected 2-year PoCE in the IVUS-guided arm is assumed to be 17%, whereas PoCE in the angiography-guided arm is assumed to be 26% [4]. It is noteworthy that the OPTIMAL trial will not use the EXCEL MI definition, but it is expected that the combined SCAI/UDMI approach will yield comparable results [37]. The cross-over rate from angio-guidance to IVUS guidance is assumed to be 4%, and the cross-over rate from IVUS guidance to angio-guidance 3%. Taking these cross-over rates into account and using a Z (standard normal) test for the difference between two proportions and a two-sided type I error of 0.05, approximately 789 750 participants are needed to demonstrate superiority of an IVUS-guided approach versus an angiography-guided one with a statistical power of 80%. A validated on-line calculator was used . A sample size of 2 x 400 399 participants allows for an attrition rate of 6%. The final sample size was set at 800. 

We added the reference to the validated tool that was used for sample size calculation, which specifically allows for incorporation of cross-over assumptions (Sealed Envelope Ltd. 2012. Power calculator for binary outcome superiority trial. [Online] Available from: https://www.sealedenvelope.com/power/binary-superiority/ and Supporting information 2). 

Please include more details on randomization (are there any blocks? How stratification is managed? Etc.).

Randomization is performed using blocks and stratification by site is performed, to ensure balance across potential local differences in treatment practices. 

Please update dates and number of patients enrolled.

We thank the Editor for this comment. By October 28th, 2021, 252 patients (32% of the target sample) have been randomised and enrolled in the OPTIMAL study. 

I would suggest to include a small final section for a brief discussion and expected results (what is expected? How the study can advance the field? Are there similar ongoing trials? Etc.)

We thank the Editor for this useful input. Echoing the extremely pertinent comment raised by Reviewer #3 we have now added a brief paragraph entitled “Clinical Perspectives” at the end of the manuscript to highlight the points suggested by the Editor. Please see pages 44-45, lines 856-872 in the “Revised Manuscript with Tracked Changes”. 

Journal Requirements:

We confirm that the manuscript meets PLOS ONE’s style requirements. We have also made changes in the structure and order of the manuscript to ensure that it aligns with the PLOS ONE sample for study protocol manuscripts: https://journals.plos.org/plosone/s/file?id=c9fb/Study%20Protocol%20Article%20Template.pdf

We confirm that the References list is complete and correct. Please note that we have added four new references: 

1) Levine GN, Bates ER, Blankenship JC, Bailey SR, Bittl JA, Cercek B, et al. 2011 ACCF/AHA/SCAI Guideline for Percutaneous Coronary Intervention: a report of the American College of Cardiology Foundation/American Heart Association Task Force on Practice Guidelines and the Society for Cardiovascular Angiography and Interventions. Circulation. 2011;124(23):e574-651. Please see the in-text citation in page 45, lines 864-866 in the “Revised Manuscript with Tracked Changes”.

2) Braunwald E. Unstable angina. A classification. Circulation. 1989;80(2):410-4. Please see the in-text citation in page 20, lines 321-323 in the “Revised Manuscript with Tracked Changes”.

3) Campeau L. Letter: Grading of angina pectoris. Circulation. 1976;54(3):522-3. Please see the in-text citation in page 20, lines 321-323 in the “Revised Manuscript with Tracked Changes”.

4) Ltd SE. Power calculator for binary outcome superiority trial. 2012 [Available from: https://www.sealedenvelope.com/power/binary-superiority/.Please see the in-text citation in page 35, line 621 in the “Revised Manuscript with Tracked Changes”.

3. Thank you for stating the following financial disclosure: "Yes. 

This is an Investigator-initiated study under the umbrella of the ECRI. Research grants have been provided by Boston Scientific and Philips Volcano to fund this study. ECRI maintains clinical trial insurance coverage for study participants in the event of trial-related injuries, if applicable and in accordance with the applicable laws and regulations of the country in which the study is performed. Also, according to the applicable regulatory requirement(s), ECRI provides insurance or indemnity (legal and financial coverage) to the Investigator/the Institution against claims arising from the trial, except for claims that arise from malpractice and/or negligence. " 

Please state what role the funders took in the study. If the funders had no role, please state: 

We confirm that “The funders had no role in study design, data collection and analysis, decision to publish, or preparation of the manuscript." We have also added this statement in our cover letter as well as in the Funding section in the text. Please see page 2, lines 29-30 in the “Revised Manuscript with Tracked Changes”.

4. Thank you for stating the following in the Competing Interests section: "Dr De Maria reports speaker fees from Miracor Medical SA. Dr Testa reports fees as medical proctor for Boston Scientific, Meril, Concept Medical, Abbott, Philips and advisory board member and/or speaker fees and/or institutional research grant from Boston Scientific, Philips, Abbott, Medtronic, Terumo, Concept Medical. Dr de la Torre Hernandez reports receipt of grants/research supports from Abbott Medical, Biosensors, Bristol Myers Squibb, Amgen and receipt of honoraria or consultation fees from Boston Scientific, Medtronic, Biotronik, Astra Zeneca, Daiichi-Sankyo. Dr Bedogni reports fees as medical proctor for BSCI, Meril, Medtronic, Terumo and advisory board member and/or speaker fees and/or institutional research grant from Boston Scientific, Philips, Abbott, Medtronic, Terumo, Concept Medical. Prof Banning reports institutional grant for fellowship form Boston and speaker fees Boston, Phillips and Miracor Medical SA. Dr Scarsini, Dr Terentes-Printzios, Dr Emfietzoglou, and Dr Spitzer have no conflict of interest to declare."

We note that you received funding from a commercial source: Boston Scientific Corporation (US)and Philips.

Please see below the amended Competing Interests Statement: 

“Dr De Maria reports speaker fees from Miracor Medical SA and research grants from Abbott and Philips. Dr Testa reports fees as medical proctor for Boston Scientific, Meril, Concept Medical, Abbott, Philips and advisory board member and/or speaker fees and/or institutional research grant from Boston Scientific, Philips, Abbott, Medtronic, Terumo, Concept Medical. Dr de la Torre Hernandez reports receipt of grants/research supports from Abbott Medical, Biosensors, Bristol Myers Squibb, Amgen and receipt of honoraria or consultation fees from Boston Scientific, Medtronic, Biotronik, Astra Zeneca, Daiichi-Sankyo. Dr Bedogni reports fees as medical proctor for BSCI, Meril, Medtronic, Terumo and advisory board member and/or speaker fees and/or institutional research grant from Boston Scientific, Philips, Abbott, Medtronic, Terumo, Concept Medical. Prof Banning reports institutional grant for fellowship form Boston and speaker fees Boston, Phillips and Miracor Medical SA. Dr Spitzer declares that the sponsor of the study is the European Cardiovascular Research Institute (ECRI), in which he is a board member, and the research organization executing the study is Cardialysis, in which he is the chief medical officer. Dr Scarsini, Dr Terentes-Printzios, Dr Emfietzoglou have no conflict of interest to declare. 

In specific relationship with the study funders:

• Boston Scientific has provided research grant support to Dr Testa, Dr Bedogni and Prof Banning

• Boston Scientific has provided speaker fees/consultancy fees/proctor fees to Dr Testa, Dr de la Torre Hernandez, Dr Bedogni and Prof Banning

• Philips has provided research grant support to Dr De Maria, Dr Testa and Dr Bedogni 

• Philips has provided speaker fees/consultancy fees/proctor fees to Dr Testa, Dr Bedogni and Prof Banning

We confirm that: “This does not alter investigators’ adherence to PLOS ONE policies on sharing data and materials.”

We have also modified the “Competing interests” section in the text. Please see pages 2-3, lines 38-63, in the “Revised Manuscript with Tracked Changes”.

5. Please amend the manuscript submission data (via Edit Submission) to include author Jose M de la Torre Hernandez MD, PhD. 

Thank you very much for spotting this. We will include author Jose M de la Torre Hernandez. 

We thank you for your suggestion. We have now moved the “Ethics approval” section from the “Declarations” section to the “Methods” section. Please see page 20, lines 310-314 in the “Revised Manuscript with Tracked Changes”.

7. We note you have included a table to which you do not refer in the text of your manuscript. Please ensure that you refer to Table 1 & 5 in your text; if accepted, production will need this reference to link the reader to the Table.

Thank you for your observation. We have modified the text. We have now moved Tables 1 and 2 and have appropriately referenced them in the text. Please see page 10, line 239 and pages 10-16 in the “Revised Manuscript with Tracked Changes”. We have also referenced Table 5 in the text. Please see page 32, line 606 in the “Revised Manuscript with Tracked Changes”.

Thank you for spotting this. We have now included a Title and a descriptive Caption for our Supporting Information File S1. We have also added a second Supporting Information File with an appropriate title and legend. Please see page 46, lines 884-892 in the “Revised Manuscript with Tracked Changes”.

Reviewers' comments:

Reviewer #1: 

1) Authors described a study protocol paper for the OPTIMAL trial. The OPTIMAL trial is a multi-center, international, randomized study to assess the superiority of IVUS-guided PCI versus qualitative angio-guided PCI in unprotected left main coronary artery disease. The manuscript is very well written. 

We thank the Reviewer for the positive feedback. 

2) I have only one comment for authors. In sample size, authors described that “The cross-over rate from angio-guidance to IVUS guidance is assumed to be 4%, and the cross-over rate from IVUS guidance to angio-guidance 3%.” I understand that the cross-over from angio-guidance to IVUS guidance would occur in some cases because of safety. However, I cannot understand why the cross-over from IVUS-guidance to angio-guidance would occur. Authors should explain why the cross-over from IVUS-guidance to angio-guidance occur in this study.

We thank the Reviewer for the pertinent comment. The possibility of cross-over from IVUS guidance to angio-guidance is indeed a less likely scenario (as highlighted by a lower anticipated cross-over rate). However, it can still occur mainly due to technical circumstances, such as vessel tortuosity or vessel angulations preventing IVUS catheter to be successfully delivered, either before or after stenting. Additionally, there are scenarios when IVUS imaging cannot be performed due to the need of rapidly completing the procedure because of clinical complications or because patient might have become unable to tolerate the procedure any longer. 

Reviewer #2: 

1) Overall, this is a comprehensive methods paper for an important and what appears to be rigorous, open-label randomised controlled trial comparing the use of IVUS to guide PCI. Whilst importance/impact may not be the journal’s priority, it is worth saying that this trial is likely to be important to the practice of coronary intervention. More specifically related to the Journal’s publication criteria, this is a detailed and rigorous trial protocol. It complies with/satisfies all of PLOS ONE’s submission guidelines and recommended considerations for study protocols.

We thank the Reviewer for the positive feeback. 

2) The authors state that they will assess angina status prior to PCI and at follow-up, but do not specify how they intend to do this. I would suggest that this is specified in the manuscript.

We thank the Reviewer for the suggestion. We agree that the method of assessing angina status should be specified in the manuscript. The patient will be contacted at follow up and will be asked about their angina status. The angina status will be established according to the Braunwald classification of unstable angina (Braunwald E et al. Circulation. 1989; 80: 410-414) and the Canadian Cardiovascular Society classification of Exertional Angina (Campeau L. et al. Circulation. 1976; 54: 522-523). This information is now explicitly stated in page 20, lines 320-322 in the “Revised Manuscript with Tracked Changes”.

Reviewer #3: 

Although one might think that it is unethical to randomize patients to undergo LMCA revascularization guided by IVUS or by angiography at present time, the fact is that clinical practice guidelines give IVUS a IIB recommendation in this setting, on the basis that there are few short randomized single-center series evaluating this issue and that most of the information has been extracted from meta-analysis of non-randomized studies. A large study including 1670 patients with left-main lesions treated with DES, (propensity score-matched analyses) showed that IVUS guidance was associated with reduced MACE within 3 years. However, MACE reduction was driven by all-cause mortality without a reduction in MI or TLR, thus leaving open the question regarding the mechanism of the observed survival benefit. Similarly, in the observational MAIN-COMPARE study, a trend for lower mortality was demonstrated, yet again without a difference in MI or TLR. In these series and meta-analysis, a repetitive reduction in mortality has been observed without a reduction of infarcts or TLR, which suggests confounding factors, since there is no mechanistic explanation for this decrease in mortality. 

Moreover, in daily clinical practice IVUS after LMCA PCI is not performed in a great percentage of procedures. Even in studies like Noble, where IVUS was strongly recommended, it was performed only in 70% of procedures. Thus, it is not uncommon that IVUS is not performed for guiding LMCA PCI, probably related to a class IIB recommendation, which perhaps is of not enough strength for a significant number of operators.

For these reasons, this reviewer considers it appropriate to carry out this randomized study, since it will definitely demonstrate without doubts whether guiding LMCA PCI by IVUS reduces MACE or not in follow-up.

We thank the Reviewer for the positive feedback. They have clearly laid out the reasons why this group of investigators has started the study. 

Major Comments:

1) Regarding Angiography-Guided Optimization, line 319. I find authors make much more emphasis in obtaining a good immediate result in IVUS guided patients than in angio guided patients. The criteria for IVUS guided optimization are quite long. However, there are only 3 lines explaining the criteria for optimal PCI of LMCA in the angio-guided. This protocol difference could account for better results in the IVUS guided group.

We thank the Reviewer for the comment. It is true that the angiography-guided optimization algorithm could appear slim compared to the IVUS-guided one. However, we are confident that the Reviewer will acknowledge and convey with us that angiography (due to its inherent limited spatial resolution and two-dimensional nature) offers only lumen diameter stenosis as a parameter to ascertain that stent deployment is optimized in terms of expansion and apposition. 

This is why the angiography-guidance algorithm could have not been made more articulated than it is and that is why it might look less detailed than the IVUS-guidance one.

2) Similarly, there are no instructions for LMCA PCI when it doesn’t involve bifurcation. POT is mandatory in bifurcation technique. But there is no recommendation of postdilation in the rest of LMCA lesions. In order to achieve the best LMCA PCI results not only in the IVUS group (which will be postdilated if needed after IVUS final interrogation) but also in the angio group, it should be mandatory to postdilate with non-compliant balloons after stent implant (to avoid stent under-expansion or malaposition which could remain unnoticed in the angio group provided IVUS won’t be done in this group)

We thank the Reviewer for the very pertinent comment. We totally agree that stent optimization and postdilations with non-compliant balloons should be recommended both in the IVUS as in the angio-guided group for LMCA bifurcation PCI as for PCI on non-LMCA or PCI on LMCA with no involvement of the bifurcation. We believe that the Reviewer will convey with us that this has now become a common practice among operators worldwide. That being said, we agree with the point raised and we have now modified the text to make this aspect clearer and more explicit in the manuscript. Please see page 23, lines 406-408, and page 24, lines 428-430 in the “Revised Manuscript with Tracked Changes”.

3) Line 250: All patients, including those already on chronic therapy with aspirin, will be preloaded with aspirin (300-325 mg) more than 2 hours prior to PCI. Pre-loading is also mandatory for all patients with a second antiplatelet agent (Clopidogrel 600 mg or Prasugrel 60 mg or Ticagrelor 180 mg) more than 2 hours prior to PCI (including patients already on chronic therapy with either of these agents). In the opinion of this reviewer this is not routine clinical practice and has no clinical justification. Moreover, it can impose a risk of bleeding in a patient if he/she receives a high loading dose of aspirin/ticagrelor/prasugrel while being chronically on this treatment

We thank the Reviewer for the comment and we totally agree with the point raised. In this regard, we need to highlight that since the initial submission of the manuscript to the Journal, a study-protocol amendment has been submitted and approved to address this aspect among others. We can confirm that the new version of the protocol does not mandate to reload a patient with an antiplatelet agent, if the patient is on chronic treatment. 

The protocol version 2.0, dated May 3rd 2021, now includes the paragraphs below: 

Aspirin. A loading dose of aspirin (150–300 mg p.o. or 75–250 mg i.v.) is mandatory if the patient is not on chronic treatment with aspirin. Either regular or chewable tablets or intravenous aspirin are used for the loading dose at least 2 hours before the procedure. A loading dose in patients on chronic treatment with aspirin is not required.

Adenosine diphosphate (ADP) antagonists. 

A loading dose of an ADP antagonist in ADP antagonist-naïve patients is mandatory. The choice of one of the following agents is left to the discretion of the Investigator. 

• clopidogrel 600 mg before PCI; or

• at sites in countries where it is approved and is commercially available, prasugrel 60 mg at least 1 hour before PCI; or ticagrelor 180 mg at least 1 hour before PCI.

For participants already receiving chronic ADP antagonist therapy, pre-loading should follow current guidelines and local practice.

The main text of the manuscript has now been amended as follows to reflect this change in the protocol. 

“All patients, unless already on chronic therapy with aspirin, will be preloaded with aspirin (300-325 mg) more than 2 hours prior to PCI. Pre-loading with a second antiplatelet agent (Clopidogrel 600 mg or Prasugrel 60 mg or Ticagrelor 180 mg) more than 2 hours prior to PCI is also mandatory unless patients are already on chronic therapy with either of these agents. For participants already receiving chronic ADP antagonist therapy, pre-loading should follow current guidelines and local practice. The choice of the second antiplatelet agent is left to the discretion of the Investigator.”

Please see pages 20-21, lines 329-335 in the “Revised Manuscript with Tracked Changes”. 

4) According to table 2 in page 6: Silent ischemia, stable angina, unstable angina or recent non-ST-segment elevation MI are inclusion criteria. However, ongoing myocardial infarction or recent myocardial infarction with rising cardiac biomarker levels are exclusion criteria. This two sentences may be confusing. A patient with recent non-ST-segment elevation MI may have rising cardiac biomarker levels.

We thank the Reviewer for this comment. Indeed, this could create confusion during enrollment, thus, in protocol V2.0 the eligibility criteria have been modified to clarify this issue. More specifically, inclusion criteria now include: “Silent ischemia, stable angina, unstable angina or non ST-segment elevation MI (NSTEMI)”; whereas “ongoing myocardial infarction or recent myocardial infarction with rising cardiac biomarker levels” has been removed from the exclusion criteria. For clarity, STEMI and cardiogenic shock remain in the exclusion criteria. 

Tables 2, 3 and 4 have now been updated accordingly in the manuscript, in order to reflect this change. Please see Table 2 page 14, Table 3 pages 17-18, and Table 4 page 18 in the “Revised Manuscript with Tracked Changes”.

5) Regarding the same aspect, in table 4 page 16 the sentence changes a bit: Ongoing myocardial infarction or recent MI with still evidence of rising cardiac biomarker levels. Please modify this sentence in table 2 in order it is clearly understood that a patient with recent non-ST-segment elevation MI can be included only if cardiac markers are already decreasing.

Please refer to our response at your previous comment (comment 4). 

6) In table 2 page 7, according to point 5, this is an exclusion criteria: “Previous history of CABG with patent graft to the LAD and/or patent graft to the LCX” However, in table 4 page 16, according to point 5 the exclusion criteria is: “Previous history of CABG with patent grafts to both LAD and LCx.” Please clarify this aspect, ¿can a patient be included if he/she has patent one graft to the LAD/LCX provided the other is occluded? According to lines 218-220 and according to table 4 it is possible, although according to table 2 page 7 it isn’t.

We thank the Reviewer for the observation. The exact wording in the protocol is “Previous history of CABG with patent graft to the LAD and/or patent graft to the LCx”. Thus, the wording in Table 2 (page 14) is the correct one. Thanks to the Reviewer, we have now also amended Table 4 to ensure consistency throughout the manuscript. Please see Table 4 page 18 in the “Revised Manuscript with Tracked Changes”. 

7) Regarding the stenting technique in LMCA-bifurcation PCI: Page 19. Line 304: Double stenting technique as bail-out after failed provisional stenting should be considered if side branch presents either 1) Angiographic DS% > 75%; 2) impaired TIMI flow (TIMI < 3); 3) ostial dissection; 4) FFR ≤ 0.80 or iFR < 0.90 or RFR < 0.90; or 5) Ostium MLA ≤ 4 mm2, with plaque burden > 50% at post-stenting IVUS on side branch (in IVUS-guided arm, only) 

• Does this mean that IVUS cannot be used in the angio-guided arm to decide if it is necessary double stent technique as bail-out? This decision must be taken only based on angiography or functional assessment though iFR, RFR or FFR in the angio-guided arm? 

• If it is like this, it should be stated that use of IVUS for this purpose in the angio-guided arm is not allowed. 

We thank the Reviewer for this useful request of clarification. We confirm that in the angiography-guided arm of the study the use of IVUS is strongly discouraged. However, the operator is allowed to use IVUS in case an imaging technique is strictly required for patient safety, i.e. in those cases showing an unclear angiographic result where there is a chance for a suboptimal result or a potential complication. We have now made this aspect more clear and explicit in the text. Please see page 24, lines 410-413 in the “Revised Manuscript with Tracked Changes”. 

Minor comments: 

8) There is a little change between DoCE and VoCE definitions

• Device-oriented Composite Endpoint (DoCE) is defined as the composite of cardiovascular death, target-vessel MI, clinically indicated repeat revascularization of the target lesion

• Vessel-oriented Composite Endpoint (VoCE) is defined as the composite of cardiovascular death, target vessel MI, repeat revascularization of the target vessel

The words “clinically indicated” may be confusing. What is important is if there was a repeat revascularization of the target lesion, not if it was clinically indicated. That words may make investigators think that if the repeat revascularization was not clinically indicated it should not be recorded.

We thank the Reviewer for these comments. DoCE and VoCE are defined very closely. DoCE follows the ARC-2 recommendations which define to the most granular but at the same time pragmatic possibility the effect of an implanted device on patient outcomes. Per ARC-2 consensus this refers to cardiovascular death (as an extension of cardiac death utilized in ARC-1), target-vessel MI (given the complexity of adjudicating target-lesion MI, which is not accepted as the most granular option for this endpoint), and target-lesion revascularization which most occur in linkage to the implanted stent (within the stented area or 5 mm proximal or distal). VoCE, however, is defined as a more inclusive endpoint that does not consider the stented lesion as the unit for a revascularization endpoint, but instead the treated vessel. 

In this regard, the concept of ‘clinically indicated’ revascularization has been consistently used following ARC-2. The protocol defines a clinically indicated Revascularization for any lesion, when the treated lesion is associated with any of the following:

• Positive invasive functional ischemia test (e.g. FFR, iFR, Doppler flow velocity reserve)

• Angiographic diameter stenosis ≥ 50% and positive non-invasive ischemia test (e.g. dobutamine stress test, nuclear test, exercise test, FFR-CT)

• Angiographic diameter stenosis ≥ 50% and ischemic symptoms (stable angina or acute coronary syndrome)

• Angiographic diameter stenosis ≥70%

• When the diameter stenosis is <50%, the presence of severe ischemic signs and symptoms (e.g. acute myocardial infarction) would also confirm the diagnosis of a clinically-indicated revascularization.

As noted by the Reviewer, this is a complex decision-making and may introduce variability in interpretation, especially in revascularizatoins of the left main especifically. Sites must report all revascularization procedures. Catheterization reports, discharge letters, and imaging data will be collected centrally and provided to the CEC (clinical events committee) for consistent classificaion of events. Thus, we expect no underreporting of revascularization events, and as for consistent classification, this will be performed by an independent CEC. 

We acknowledge the perspectives of the Reviewer, but respectfully wish to maintain the definition of DoCE as is, since it is consistent with the ARC-2 principles. Nevertheless, this is indeed a special point of attention at the adjudication process.

---

## [Editor Report · Decision Letter 1]

17 Nov 2021

A multi-center, international, randomized, 2-year, parallel-group study to assess the superiority of IVUS-guided PCI versus qualitative angio-guided PCI in unprotected left main coronary artery (ULMCA) disease: Study protocol for OPTIMAL trial

PONE-D-21-13023R1

Dear Dr. Banning,

We’re pleased to inform you that your manuscript has been judged scientifically suitable for publication and will be formally accepted for publication once it meets all outstanding technical requirements.

Kind regards,

Giuseppe Gargiulo, MD, PhD

Academic Editor

PLOS ONE

Additional Editor Comments (optional):

The authors have properly addressed all minor requests and the paper is improved.
---

## [Editor Report · Acceptance letter]

26 Dec 2021

PONE-D-21-13023R1 

A multi-center, international, randomized, 2-year, parallel-group study to assess the superiority of IVUS-guided PCI versus qualitative angio-guided PCI in unprotected left main coronary artery (ULMCA) disease: Study protocol for OPTIMAL trial 

Dear Dr. Banning:

I'm pleased to inform you that your manuscript has been deemed suitable for publication in PLOS ONE. Congratulations! Your manuscript is now with our production department. 

Kind regards, 

on behalf of

Dr. Giuseppe Gargiulo 

Academic Editor

PLOS ONE